# Tuning dCas9's ability to block transcription enables robust, noiseless knockdown of bacterial genes

Antoine Vigouroux[1,2], Enno Oldewurtel[2] , Lun Cui[1] , David Bikard[1,*] & Sven van Teeffelen[2,**]

## Abstract

Over the past few years, tools that make use of the Cas9 nuclease have led to many breakthroughs, including in the control of gene expression. The catalytically dead variant of Cas9 known as dCas9 can be guided by small RNAs to block transcription of target genes, in a strategy also known as CRISPRi. Here, we reveal that the level of complementarity between the guide RNA and the target controls the rate at which RNA polymerase "kicks out" dCas9 from the target and completes transcription. We use this mechanism to precisely and robustly reduce gene expression by defined relative amounts. Alternatively, tuning repression by changing dCas9 concentration is noisy and promoter-strength dependent. We demonstrate broad applicability of this method to the study of genetic regulation and cellular physiology. First, we characterize feedback strength of a model auto-repressor. Second, we study the impact of amount variations of cell-wall synthesizing enzymes on cell morphology. Finally, we multiplex the system to obtain any combination of fractional repression of two genes.

**Keywords** CRISPR-dCas9; CRISPRi; gene-expression noise; peptidoglycan cell wall; single-cell

**Subject Categories** Quantitative Biology & Dynamical Systems; Synthetic Biology & Biotechnology; Transcription

**Mol Syst Biol. (2018) 14: e7899**

## Introduction

A powerful way to investigate genes and their regulation in bacteria is to vary their expression levels and investigate the response of the cell. To that end, genes are typically placed under inducible promoters. While easy to implement, this approach has multiple disadvantages: First, native expression can lie outside the dynamic range of the inducible promoter. Second, inducible promoters typically increase expression noise in comparison with native promoters

(Elowitz *et al*, 2002). And third, only few orthogonal inducible systems exist, thus making multiplexing difficult. Recently, different strategies have been devised to knock down gene expression by relative amounts from their native levels: Specifically, antisense transcription can reduce gene expression in a defined manner (Brophy & Voigt, 2016). While this approach works well for moderate promoter strength, it becomes less efficient the stronger the promoter. As an alternative strategy, genes can be knocked down from their native locus to varying degrees using CRISPR technology (Bikard *et al*, 2013; Qi *et al*, 2013). The catalytic mutant form of the RNA-guided Cas9 nuclease from *Streptococcus pyogenes* (dCas9) can be easily programmed to bind any position of interest on the chromosome, with the requirement of an "NGG" protospacer adjacent motif (PAM). dCas9 is unable to cleave target DNA, but still binds DNA strongly. If the target is chosen downstream of the promoter, dCas9 serves as a roadblock that blocks transcription elongation. Here, we characterize this system at the single-cell level, with interesting implications for the native CRISPR immune system. We then develop a strategy to use this system for precise and noise-preserving relative gene repression that is independent of promoter strength.

Target search of Cas9 begins by probing DNA for the presence of a PAM motif followed by DNA melting and complementarity-dependent RNA strand invasion (Sternberg *et al*, 2014; Szczelkun *et al*, 2014). While complementarity in the PAM-proximal region known as the seed sequence is important for binding, several mismatches in the PAM-distal region can be tolerated as demonstrated by DNA binding assays (Kuscu *et al*, 2014; Wu *et al*, 2014) and by monitoring target-gene repression in *Eschericha coli* (Bikard *et al*, 2013). The degree of gene repression can then be controlled quantitatively in two different ways: first, by changing the level of dCas9 expression from an inducible promoter, which impacts the probability of dCas9 binding to target DNA. This has recently been demonstrated in *Bacillus subtilis* where dCas9 was placed under the control of a xylose-inducible promoter (Peters *et al*, 2016), as well as in an *E. coli* strain modified to enable tunable control of expression from a $P_{BAD}$ promoter (Li *et al*, 2016); second, by introducing mismatches between the guide RNA and the target DNA, as demonstrated in *E. coli* (Bikard *et al*, 2013). While a perfectly matched guide RNA leads

---

1 Synthetic Biology Laboratory, Institut Pasteur, Paris, France
2 Microbial Morphogenesis and Growth Laboratory, Institut Pasteur, Paris, France
 *Corresponding author. Tel: +33 1 45 61 39 24; E-mail: david.bikard@pasteur.fr
 **Corresponding author. Tel: +33 1 45 68 80 16; E-mail: sven.van-teeffelen@pasteur.fr

---

to very strong repression, decreasing complementarity in the PAM-distal region progressively reduces the repression strength (Bikard *et al*, 2013).

Here, we compare these two repression strategies by characterizing the properties of dCas9-mediated repression at the single-cell level. This enables us to propose a novel physical model of dCas9-mediated repression. It was previously assumed that decreased levels of guide RNA complementarity would decrease repression strength by virtue of reduced occupancy of the target by dCas9 (Farasat & Salis, 2016). Here, we demonstrate a different mechanism: If the target is inside an open reading frame (ORF), complementarity determines the probability that RNA polymerase (RNAP) kicks out dCas9 during the transcription attempt, while the rate of spontaneous dCas9 unbinding is negligibly small. If dCas9 levels are high enough to saturate the target, this mechanism alone determines repression strength. This leads to desirable properties: first, relative repression strength is independent of native expression levels. Second, repression does not add any extrinsic noise to gene expression. On the contrary, tuning gene expression by changing the level of dCas9 expression is inherently noisy and depends on the promoter strength of the target.

We demonstrate the use of complementarity-based CRISPR knockdown in combination with fluorescent-protein reporters inserted upstream of a gene of interest to precisely and robustly control its expression. The use of reporter gene fusions rather than direct targeting of the gene of interest yields a predictable repression fold as characterized in this study and provides an easy way to monitor expression levels in single cells. We demonstrate the versatility of our approach using two examples: first, the accurate control of the rate at which the RNAP kicks out dCas9 enables us to quantify the degree of feedback in a model auto-repressor by measuring how much actual gene expression differs from the controlled rate. Second, we take advantage of the ability to obtain a precise degree of repression during steady-state growth to investigate the impact of expression level of an operon coding for two essential cell-wall synthesis enzymes of the "rod" complex, PBP2 and RodA. Finally, we demonstrate that this system can be easily and robustly multiplexed to obtain any combination of the fractional repression of two genes.

## Results

### Varying levels of guide RNA-target complementarity enables controlling gene expression without addition of noise

To quantify how CRISPR-dCas9 modulates gene expression at the single-cell level, we integrated expression cassettes for two constitutively expressed reporters, *sfgfp* coding for the superfolder green fluorescent protein (GFP) and *mCherry* coding for a red fluorescent protein (RFP) at two different chromosomal loci of *E. coli* strain MG1655. To repress either of these genes using CRISPR knock-down, we integrated the *dcas9* gene from *S. pyogenes* under a $P_{tet}$ promoter, inducible by the addition of anhydrotetracycline (aTc) (Qi *et al*, 2013). We then guided the dCas9 protein to target the coding strand of GFP- and RFP-coding ORFs using a constitutively expressed CRISPR array coding for the guide RNAs and the necessary tracrRNA, which form a complex together with dCas9 (Hsu

*et al*, 2014). Inducing dCas9 expression in this setup did not have an impact on growth (Appendix Fig S1). We also measured the stability of the target-gene repression over time and saw repression over 5 days of culture. Once we stopped dCas9 induction all 40 clones tested recovered the target-gene expression. This genetic system is thus very stable, and dCas9 expression did not show any toxicity.

In this system, repression strength can be tuned in two different ways: either by modulating dCas9 expression level using different aTc concentrations or by modulating spacer complementarity to the target gene using different numbers of mismatches at the 5′ side of the spacer. We employed these two different strategies to repress GFP by different amounts and measured GFP concentration at the single-cell level by high-throughput microscopy (Fig 1A and B). As expected, average GFP levels decreased with increasing aTc concentration or increasing spacer complementarity. However, the distributions of single-cell GFP concentrations differed significantly between the two different modes of repression modulation (Fig 1C and D). Specifically, using a perfectly matched guide RNA and varying aTc concentrations led to large cell-to-cell fluctuations in the intermediate induction regime, where fluctuations of dCas9 levels strongly affect gene expression (Fig 1D and Appendix Fig S2). When dCas9 was not induced, fluctuations of the non-repressed constitutive promoter were recovered. For strong dCas9 expression, fluctuations were presumably reduced as the target site was saturated with dCas9 as elaborated below. On the contrary, inducing dCas9 at a constant high level with 100 ng/ml of aTc and varying the degree of guide RNA complementarity maintained the noise (standard deviation over the mean) of single-cell GFP concentration almost constant (Fig 1C and Appendix Fig S2). The plateau value of the expression noise of about 0.3 (corresponding to cell-to-cell variations of 30%) is similar to measurements made by others for constitutive genes in wild-type *E. coli* (Taniguchi *et al*, 2010). Complementarity-based gene repression is qualitatively different from gene repression using transcriptional repressors. For example, the Lac repressor can increase the extrinsic part of the noise of its targets by about fivefold as compared to the unrepressed case (see Appendix Fig S3), in agreement with previous measurements (Elowitz *et al*, 2002). Accordingly, a similar increase of noise is observed if repression is modulated by inducer concentration. The alternative system proposed here thus enables to tune expression levels with high precision in single cells.

### RNAP can transcribe dCas9-bound targets in a complementarity-dependent manner

The lack of additional noise in gene expression at high dCas9 concentrations for different numbers of mismatches suggested to us that repression might be independent of fluctuations in dCas9 concentration. To test this hypothesis, we reduced the fraction of active dCas9 complexes roughly by a factor of two by introducing a decoy guide RNA (Fig 2A). We then measured population-averaged gene expression by flow cytometry. Indeed, we found the level of gene repression to be constant in the presence or absence of the decoy for both high and low degrees of complementarity (Fig 2B), confirming the hypothesis that the target is saturated by dCas9 for degrees of complementarity between 20

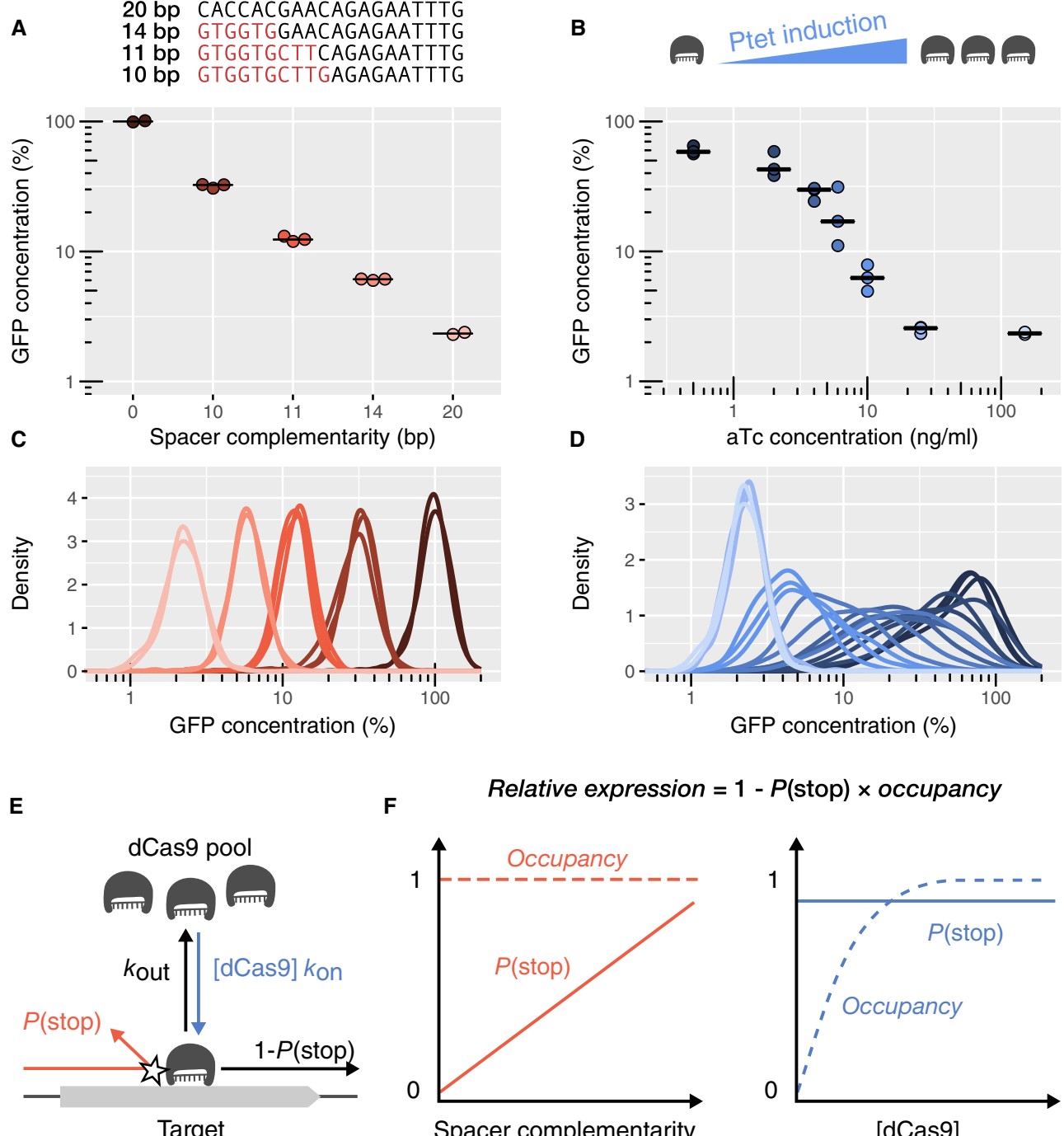

**Figure 1.  In saturating conditions, CRISPR knockdown can modulate gene expression over a large dynamic range without generating noise.**

A, B     Average cellular GFP concentration obtained (A) by changing guide RNA-target complementarity at a constant high dCas9 concentration or (B) by varying dCas9 levels with increasing concentration of the aTc inducer. Relative GFP concentrations are obtained by high-throughput microscopy and given relatively to the non-targeting spacer at high dCas9 expression. Individual points represent independent replicates. Horizontal bars represent the median of three replicates.

C, D     Distribution of GFP concentrations for each experiment in panels (A and B). Curves of the same color represent replicates of the same condition.

E        Mechanistic model of dCas9-mediated repression. The expression level of a dCas9-targeted gene is reduced by the product of two probabilities: the probability $P$(stop) of dCas9 blocking RNAP upon collision if occupying the target, and the probability of dCas9 occupying the target (termed occupancy). The occupancy is determined by binding constant $k_{on}$, dCas9 concentration [dCas9], and dCas9 unbinding rate $k_{out}$. The unbinding rate $k_{out}$, in turn, is the sum of transcription-independent unbinding rate and kick-out rate due to collision with the RNAP (see Materials and Methods for details).

F        The two panels schematically illustrate the behavior of the probability of dCas9 blocking RNAP $P$(stop) and dCas9 occupancy if repression strength is controlled by guide RNA complementarity (left) or dCas9 concentration (right), respectively.

Source data are available online for this figure.

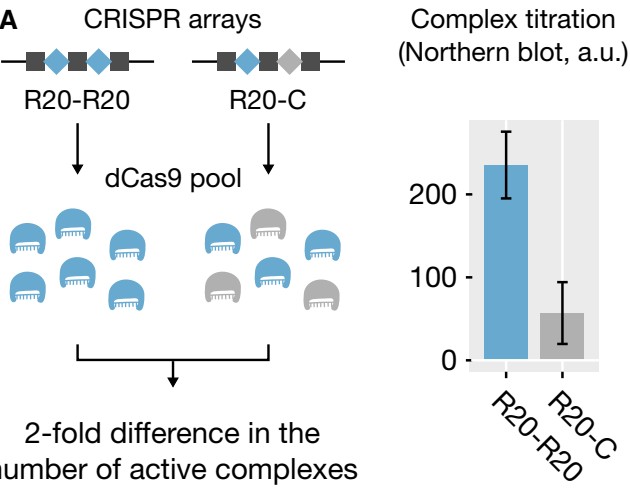

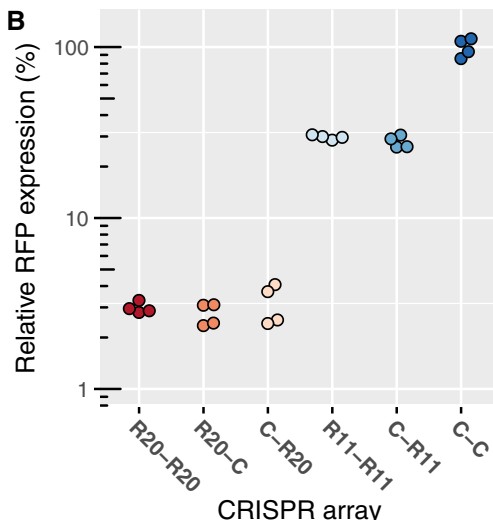

**Figure 2.  In saturating conditions, CRISPR knockdown by mismatched guide RNAs does not depend on the concentration of active dCas9 complexes.**

A    Left: Schematic of the assay used to investigate dependence on dCas9 complex concentration. R20 is a spacer targeting RFP with a perfect match. R11 targets RFP with 11 bp of complementarity. C is a non-targeting spacer. Introducing the spacer C in the CRISPR array acts as a decoy and halves the concentration of active dCas9 complex. Right: Northern blot measurement of the concentration of the processed guide RNA R20, reflecting the amount of complexes carrying R20 at the moment of the measurement. Error bars represent standard deviations of three biological replicates. a.u.: arbitrary units.

B    Flow cytometry measurement of relative RFP expression levels, with each point representing one biological replicate. The values are normalized with respect to the non-targeting CRISPR array (C-C). Expression did not differ in the presence of the decoy (C-R20 vs. R20-R20, P-value: 0.68), nor when the order of the array was reversed (C-R20 vs. R20-C, P-value: 0.21), even with only 11 bp of complementarity (C-R11 vs. R11-R11, P-value: 0.53). P-values come from a two-sided Student's t-test applied to the natural logarithms of the mean expression (significance threshold: 0.017 after Bonferroni correction).

Source data are available online for this figure.

and 11 bp. This remained true even with three decoy spacers in a CRISPR array, regardless of the position of the active spacer in the array (Appendix Fig S4). The effectiveness of the decoy strategy was confirmed by gradually lowering the concentration of aTc until we observed the transition from strong repression to no repression. As expected, the transition happened at higher aTc concentrations with three decoys than with one (Appendix Fig S4B), confirming that decoys reduce the concentration of active complex. In both cases, at high induction, the residual expression reached a plateau value around 3%, corresponding to the concentration-independent regime. We note that these and the following measurements of population averages are performed by flow cytometry and are thus generally noisier than the results obtained by high-throughput microscopy presented in Fig 1.

Previously, it was thought that the repression strength due to dCas9 is solely determined by the occupancy of target DNA, that is, by the rates of target binding and spontaneous unbinding. According to this simple view, low and intermediate levels of target repression should inherently depend on dCas9 concentration, as higher dCas9 concentrations lead to higher equilibrium binding rates and thus higher occupancy, if the target is not fully occupied. This view is in clear contradiction to the observed independence of repression on dCas9 concentration for low and intermediate levels of repression (Fig 2B). On the contrary, independence of dCas9 concentration suggests that the target is saturated by dCas9, that is, that dCas9 is bound to the target at almost all times, and that a different mechanism must be responsible for different degrees of repression strength.

To reconcile the robustness of repression strength with respect to dCas9 concentration, we hypothesize that residual expression of the target gene might be possible even if dCas9 is saturating the target. We suggest that upon collision of RNAP with dCas9, dCas9 blocks the RNAP with a probability $P(\text{stop}) \leq 1$ that depends on guide RNA-target complementarity (Fig 1E). If $P(\text{stop}) = 1$, the system efficiently blocks RNAP every time RNAP and dCas9 collide. At the opposite extreme, if $P(\text{stop}) = 0$, dCas9 never blocks RNAP (Fig 1F). According to this mechanism, the expression level of a dCas9-targeted gene is given by

$$\gamma = \gamma_0[1 - P(\text{stop})P(\text{bound})].$$

Here, $\gamma_0$ is the native transcription rate and $P(\text{bound})$ is the probability that dCas9 is occupying the target.

The probability $P(\text{stop})$ only depends on guide RNA-target complementarity. Therefore, repression is independent of dCas9 concentration, if the occupancy is very close to 1, that is, if the target is saturated. In these conditions, cell-to-cell fluctuations of dCas9 concentration also no longer affect the repression of the target, thus explaining the low and constant noise obtained for different degrees of repression (Fig 1C and Appendix Fig S2).

Interestingly, when the same target is moved from the ORF to the promoter region, repression is increased and depends on concentration of active dCas9 complex (Appendix Fig S5). This finding suggests that RNAP can pass the occupied target site inside the ORF thanks to its processive polymerase activity, but that the RNAP cannot bind at the occupied target site inside the promoter region, where it relies on diffusion.

### If dCas9 is saturating the target, relative repression is independent of target-gene promoter strength

To use CRISPR knockdown on genes with different native expression levels, it is important to know whether the transcription rate of the target has an influence on the relative repression. According to our model definition, the probability $P$(stop) that dCas9 blocks RNAP does not depend on promoter strength. Repression strength should thus not be measurably affected for promoters of different strengths, if dCas9 is saturating the target, that is if $P$(bound) is very close to 1. To verify this prediction, we put *sfgfp* under the control of two promoters of different strengths ($P_{127}$ and $P_{PhlF}$) and blocked expression using four different guide RNAs with an increasing number of mismatches. While the strain with $P_{PhlF}$ expressed about three times more GFP than the strain with $P_{127}$ (Fig 3A), the repression fold with regard to the promoter's initial expression level was identical in each case (Fig 3B). We found the same behavior when we compared $P_{127}$ with the 12 times weaker $P_{Lac}$ promoter with 1 mM IPTG (Appendix Fig S6). These observations confirm that repression by mismatched guide RNAs in saturating conditions is independent of promoter strength.

### If dCas9 is not saturating the target, relative repression depends on promoter strength, supporting a "kick-out" model of dCas9 ejection by RNAP

As transcription can be successful if dCas9 is saturating the target, we wondered whether dCas9 would be ejected from the target by RNAP during successful transcription events. While physical displacements would not affect repression in saturating conditions, they could measurably reduce the occupancy of the target $P$(bound)

if dCas9 does not saturate the target, for example, if repression is controlled by dCas9 concentration. According to our kick-out model of dCas9 ejection by RNAP, the occupancy is given by

$$P(\text{bound}) = \frac{k_{\text{on}}[\text{dCas9}]}{k_{\text{on}}[\text{dCas9}] + k_{\text{out}}}.$$

Here, $k_{\text{on}}[\text{dCas9}]$ is the rate of binding and $k_{\text{out}}$ is the combined rate of RNAP-induced ejections, spontaneous unbinding and possibly replication-fork-based displacements (Jones *et al*, 2017; see Materials and Methods for details). A stronger promoter would increase the unbinding rate $k_{\text{out}}$ and therefore reduce the occupancy $P$(bound), which, in turn, would reduce the repression fold. Indeed, we observed a weaker repression of the stronger $P_{PhlF}$ promoter compared to $P_{127}$ at low dCas9 concentrations. This observation quantitatively agrees with our kick-out model for full and intermediate (14 bp) levels of complementarity, respectively (Fig 3C and Appendix Fig S7A). For these levels of complementarity, our model also predicts that unbinding is dominated by kick-out events, while spontaneous unbinding is rare (Materials and Methods; Appendix Fig S7B).

For full complementarity, our model is compatible with the hypothesis that dCas9 never leaves the target spontaneously but gets kicked out either by the RNAP or during DNA replication. This prediction is consistent with the long half-life of dCas9 binding recently reported *in vivo* (Jones *et al*, 2017) and previously reported *in vitro* (Sternberg *et al*, 2014).

The kick-out model is expected to be valid for levels of complementarity lower than 14 bp, as the rate of successful transcription is increased (Fig 1A) while the target remains saturated down to 11 bp at high dCas9 concentrations (Fig 2B). However, spontaneous

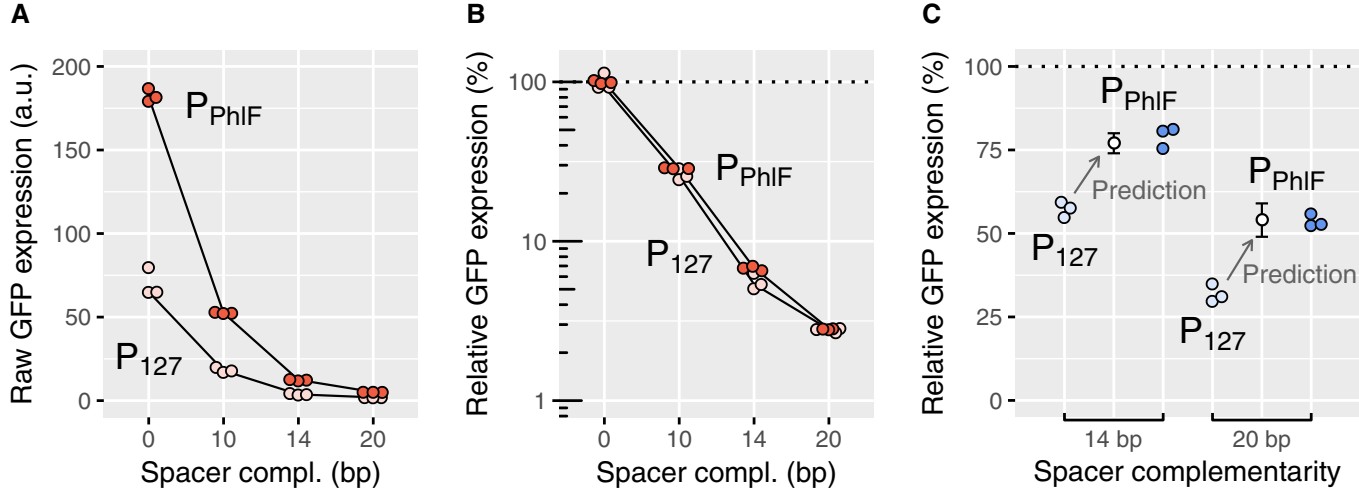

**Figure 3. Relative repression by dCas9 is independent of promoter strength only in saturating conditions.**

Relative GFP expression measured by flow cytometry for two promoters of different strengths ($P_{127}$ and $P_{PhlF}$) and repressed using the same set of spacers for saturating (A, B) and non-saturating (C) dCas9 concentrations.

A, B    Raw GFP expression (A) and relative GFP expression with respect to a non-targeting spacer (B) for a saturating dCas9 concentration. While $P_{PhlF}$ is about three times stronger than $P_{127}$, the relative expression levels after repression are similar for both promoters.

C    Experimental and predicted relative GFP expression for a non-saturating dCas9 concentration (using a 40 times lower concentration of aTc). Repression is weaker for the stronger $P_{PhlF}$ promoter for up to six mismatches on the guide RNA, in quantitative agreement with the kick-out model (see Appendix). Error bars: standard error of the mean of the computational prediction.

Source data are available online for this figure.

unbinding is expected to become equally or more important than collision-based ejections below some level of complementarity below 14 bp. Yet, at high dCas9 concentrations used for all applications below, the combined rates of unbinding and ejections are still much lower than the rate of rebinding (see previous paragraph).

Finally, we note that our observation of promoter-strength dependence is compatible with any mechanism, for which the ejection rate is proportional to transcription rate, that is, it is in principle possible that a fraction of successful transcription events leaves dCas9 bound to the coding strand while the RNAP reads the template strand (see Materials and Methods for details).

### dCas9 ejection probability increases with temperature

It was recently reported (Wiktor *et al*, 2016) that dCas9 is no longer active at 42°C, suggesting that repression strength might decrease with increasing temperature. This observation also bears the possibility that our system becomes less robust with respect to dCas9-copy number fluctuations and promoter strength with increasing temperature, if the condition of target saturation was not fulfilled. To quantify the temperature dependence of repression and test for robustness, we measured the repression of RFP by guide RNAs with 11 bp or 20 bp of complementarity at temperatures ranging from 30 to 42°C. The repression strength decayed continuously with increasing temperature (Fig 4), displaying a sharp decrease of repression between 37 and 42°C. Regardless of the temperature, repression strength was not affected by dCas9 complex concentration (Appendix Fig S8). From our model, we can thus conclude that increasing temperature does not affect dCas9 occupancy but increases the probability of dCas9 being kicked out by the RNAP. This also indicates that our system should work independently of promoter strength at all temperatures tested.

### CRISPR knockdown in combination with fluorescent-protein insertions can be used to repress and monitor genes in their native contexts

Precision, robustness, and large dynamic range make complementarity-based CRISPR knockdown a versatile repression strategy. To repress genes of interest in their native context, we propose to insert *sfgfp* or *mCherry* reporters as transcriptional or translational fusions upstream of the gene. We provide here a convenient CRISPR-based method to perform these insertions inspired by a previous allelic exchange strategy (Pósfai *et al*, 1999) (see Appendix Text and Appendix Fig S9). A library of CRISPR plasmids can then be introduced to repress the fusions to the desired levels by targeting the *sfgfp* or *mCherry* coding sequences. The method thus allows taking advantage of the measured repression levels for constitutive promoters established above. Furthermore, gene expression can be measured at the single-cell level, revealing cell-to-cell variations. The library of CRISPR plasmids used here can be obtained through addgene (https://www.addgene.org/depositor-collections/bikard-crispr-repression/).

In the following, we demonstrate that this system has broad applicability for the study of genetic regulation and cellular physiology: first, we study the regulation of a model transcriptional feedback circuit, and second, we quantify the effect of fractional protein repression on cell morphology during steady-state growth.

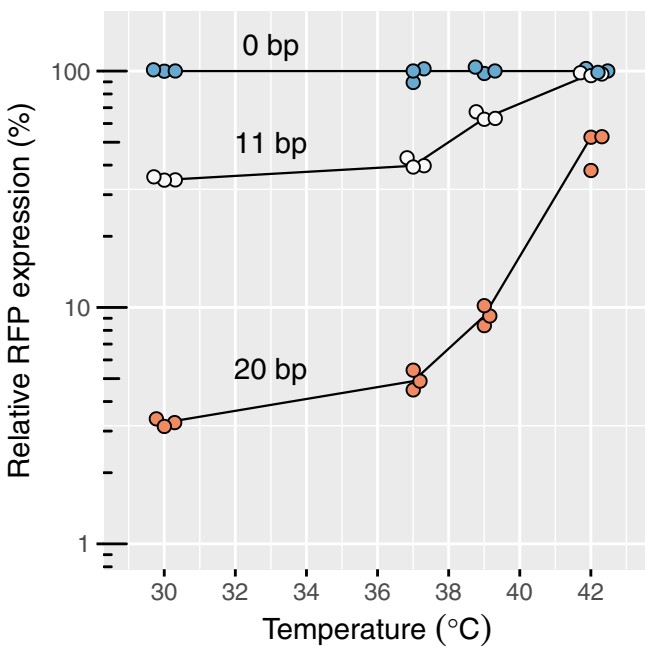

**Figure 4. The efficiency of CRISPR knockdown is affected by high temperatures.**

Relative RFP expression measured by flow cytometry upon repression with different levels of complementarity and at different temperatures. The values are normalized with respect to the non-targeting spacer at each temperature.

Source data are available online for this figure.

### CRISPR knockdown can be used to uncover and characterize genetic feedback

To demonstrate the versatility of our system for the study of genetic circuits, we chose the previously described PhlF auto-repressor from *Pseudomonas fluorescens* (Abbas *et al*, 2002) as a model system: We constructed a synthetic operon consisting in the $P_{PhlF}$ promoter followed by the *sfgfp* and *phlF* genes in a single operon (Fig 5A). The PhlF repressor binds to the $P_{PhlF}$ promoter and decreases transcription initiation, thus creating an artificial negative feedback loop. The strength of this feedback can be externally reduced by adding the chemical inducer 2,4-di-acetyl-phloroglucinol (DAPG) that blocks binding of PhlF to the promoter. Accordingly, higher DAPG concentrations lead to higher steady-state concentrations of PhlF and GFP (Appendix Fig S10). To determine whether PhlF binds to the operator cooperatively, we aimed to quantify the feedback strength as a function of promoter strength for different DAPG concentrations. To mimic different promoter strengths, we targeted the *sfgfp* ORF using spacers with variable degrees of complementarity (Fig 5). CRISPR knockdown of GFP should lead to an increased transcription-initiation rate of the promoter. As a consequence, the fold change of expression during CRISPR knockdown should be lower in the case of feedback than without feedback. The quantitative difference between the two situations can then be used to quantify the feedback strength.

As anticipated, expression of GFP decreased with increasing complementarity and the relative reduction of expression was less

pronounced with feedback than without feedback (Fig 5B). We then fit the expression data to a mathematical model of gene repression (Fig 5B, Appendix Fig S11 and Appendix Text) to calculate for each DAPG concentration the binding constant of the repressor $K$ and a Hill coefficient $n$, which describe the dependence of repression on promoter strength. We observe that a Hill coefficient of $n = 2$ describes our data for low DAPG concentrations (0 and 5 μM), while a Hill coefficient of $n = 1$ was required to describe our observations at 50 μM. PhlF proteins dimerize *in vitro* and are thought to bind the operator as a dimer (Abbas *et al*, 2002). To reconcile our observation, we speculated that PhlF might be predominantly found as monomers at high DAPG concentrations and as dimers at low DAPG concentrations (see the Appendix Text for details). However, the detailed mechanism underlying the sharp transition in Hill coefficients remains to be studied by independent experiments.

The detailed insights obtained here demonstrate the usefulness of precisely controlling the rate at which the RNAP is blocked by dCas9, while monitoring residual expression with a fluorescent reporter. The same method can be applied to other and more complex problems of gene regulation, for example, by monitoring the response of one gene to the precisely tuned levels of another gene repressed by CRISPR knockdown.

## CRISPR knockdown reveals how cells adapt their shapes to low levels of an essential cell-wall synthesis operon

We then used our approach to explore the morphological response of cells to different expression levels of two essential proteins for peptidoglycan cell-wall synthesis encoded by the *mrdAB* operon. PBP2 (encoded by *mrdA*) and RodA (encoded by *mrdB*) are inner membrane proteins with, respectively, transglycosylase (Meeske *et al*, 2016) and transpeptidase activity (Sauvage *et al*, 2008). The two highly conserved enzymes are part of the multi-enzyme "rod" complex, which is essential for cell-wall synthesis during cell elongation (Cho *et al*, 2016).

Previous depletion experiments suggest that PBP2 expression is buffered against large fluctuations in enzyme number, as cells grow for multiple generations before showing a reduction of growth rate (Lee *et al*, 2014). The drawback of depletion experiments is that they do not allow studying the effect of protein abundance in the steady state. To quantify the relation between PBP2 levels and morphological response during steady-state growth, we constructed a translational protein fusion by seamlessly integrating *mCherry* in front of the *mrdA* ORF in the native chromosomal *mrdAB* locus (Fig 6A). The mCherry-PBP2 fusion is fully functional, similarly to a fusion constructed previously (Lee *et al*, 2014). We then introduced a chromosomal P$_{tet}$-*dCas9* cassette and different pCRRNA plasmids programmed to target *mCherry* with 0, 11, 18 or 20 bp of complementarity in order to obtain a range of transcription rates for the operon. These strains were induced for dCas9 expression and grown until protein levels and cell dimensions reached steady state (Appendix Fig S12). Single-cell measurements were then performed by phase-contrast and epi-fluorescence microscopy.

Lowering expression of the *mrdAB* operon led to increasing cell width, with a sharp rise of cell width below ~20% of the native expression level (Fig 6B and C, and Appendix Fig S13A), while cell length was largely unaffected except for the highest repression

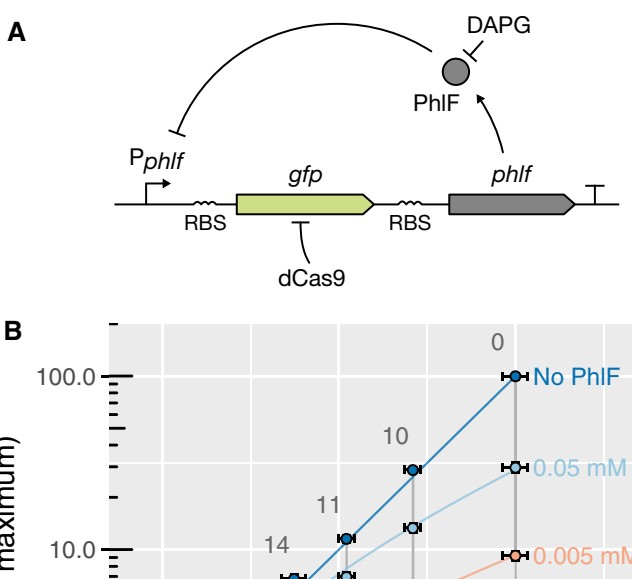

**Figure 5.  CRISPR knockdown can be used to quantitatively characterize feedback loops.**

A   Schematic of the synthetic feedback loop constructed for this experiment. The strength of the feedback can be modulated by addition of DAPG, an inhibitor of PhlF. RBS: ribosome binding site. T: transcription terminator.

B   Flow cytometry measurements and fits to a theoretical model of relative GFP expression levels, where GFP is expressed from the artificial feedback loop presented in panel (A). GFP expression is normalized by the maximal level of GFP expressed constitutively from the P$_{PhlF}$ promoter alone (indicated as "No PhlF"). The GFP is repressed using four different guide RNAs with, respectively, 10, 11, 14, and 20 bp of complementarity. The passage probability $1 - P$(stop) associated with each of these guide RNAs was measured in parallel on a strain expressing GFP constitutively from the P$_{127}$ promoter. Adding different amounts of DAPG to the medium reduces the strength of the feedback, causing the steady-state level to increase and repression to become more efficient. The colored lines represent the GFP expression as predicted by a mathematical model that was fitted to the data (see Appendix). For each DAPG concentration, a binding constant characterizing the strength of the feedback and a Hill coefficient were determined. Error bars: 95% confidence interval of the mean based on three biological replicates.

Source data are available online for this figure.

strength (Appendix Figs S13B and S14), consistently with PBP2 and RodA being essential for building the cylindrical part of the cell wall but not the cell septum. We then wondered whether enzyme levels in individual cells were responsible for cell-to-cell variations in cell

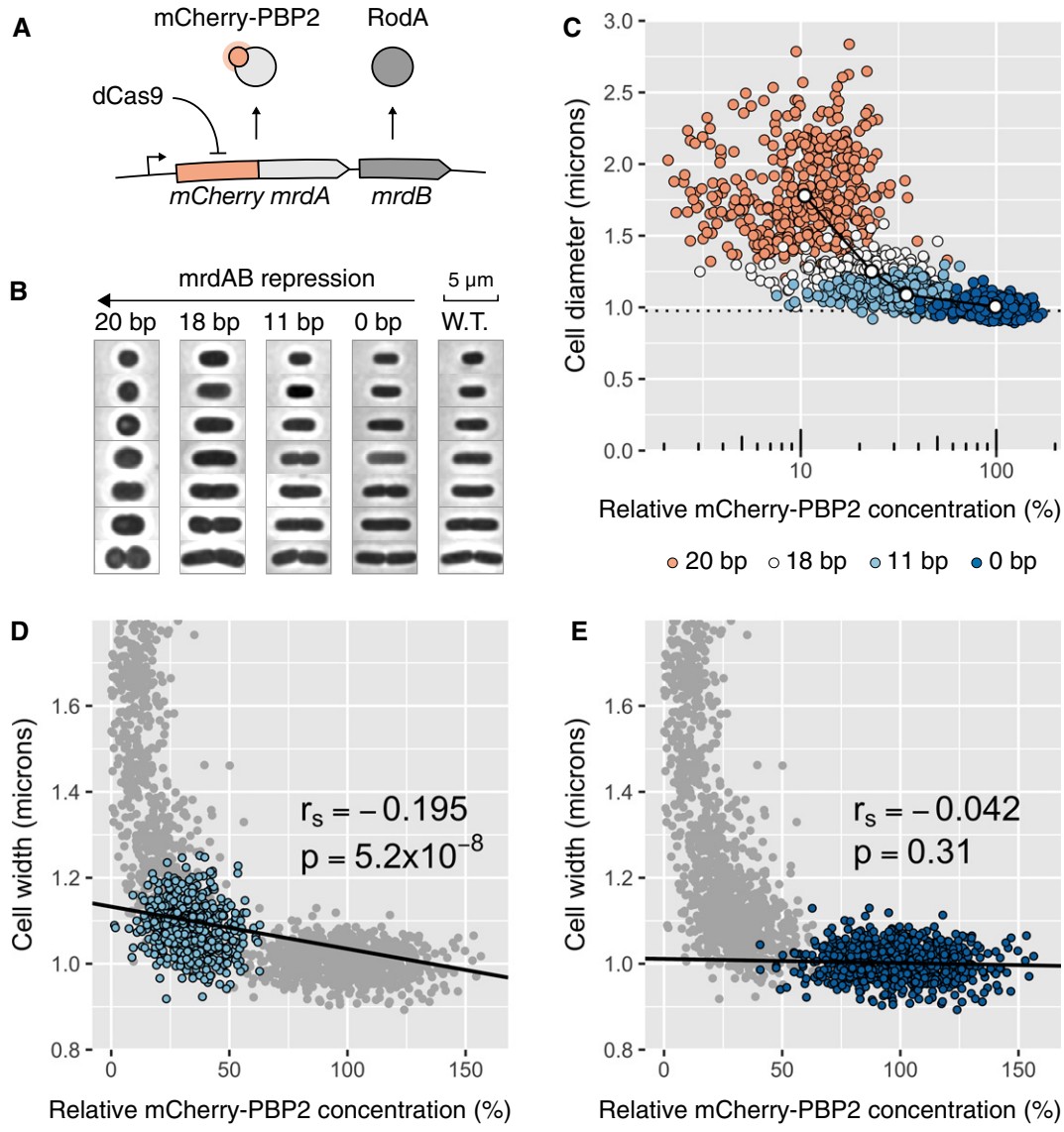

**Figure 6. CRISPR knockdown of the *mrdAB* operon increases cell width at high repression strengths.**

A   Schematic of the modified chromosomal locus of the *mrdAB* operon in strain AV08.

B   Cell shapes observed by phase-contrast microscopy for cells grown in M63 minimal medium. Different repression levels of the *mrdAB* operon are compared to wild-type *Eschericha coli*. Cells with different cell lengths were picked at random and images were rotated numerically.

C   Cell width as a function of the mCherry-PBP2 concentration measured by fluorescence microscopy. Each point represents a cell, and colors represent different levels of spacer complementarity. The connected white dots represent the population averages (mean of three biological replicates). The dotted line represents the average cell width for wild-type *E. coli* (mean of three replicates). The values are normalized with respect to the non-targeting spacer.

D, E   Linear regression between mCherry-PBP2 concentration level and cell width, for the strains repressed with 11 bp (panel D) and 0 bp (no repression, panel E). $r_S$ is the Spearman correlation coefficient (median of three biological replicates). The negative value indicates that cells with a lower level of PBP2/RodA tend to be wider. The *P*-values (two-sided *F*-test) measure the certainty that the slope is different from 0.

Source data are available online for this figure.

diameter at low or intermediate expression levels, where the average cell diameter was affected by *mrdAB* repression. Indeed, we found cell-to-cell fluctuations in the intracellular density of mCherry-PBP2 to be negatively correlated with cell diameter for intermediate *mrdAB* repression (at 11-bp guide RNA/target complementarity; Fig 6D). Such a correlation was not observed when the operon was not repressed (Fig 6E), indicating that the cells buffer

natural fluctuations of *mrdAB* and thus avoid fluctuations of cell morphology, as previously suggested (Lee *et al*, 2014). By gradually lowering the levels of PBP2 and RodA, we were able to take the cells out of the buffering regime at about 30% of native expression. Together, these experiments demonstrate that cells buffer stochastic gene expression of an essential operon against fluctuations of about threefold and that cells cope with even stronger fluctuations by

adjusting their surface-to-volume ratio. However, once expression levels are reduced by more than fivefold, cells show severe growth defects.

**CRISPR knockdown can be used to modulate the expression of two genes**

With our method, the fractional repression level of any target gene is controlled genetically rather than chemically by the concentration of an inducer. It can thus be used to modify expression of multiple genes independently. To assess this potential, we built a library of CRISPR arrays containing two spacers, one targeting *sfgfp* and the other *mCherry*. We selected five spacers with varying levels of complementarity to each of the target, spanning a large range of expression, from 2 to 100% of the initial level. We combined these spacers to form 20 CRISPR arrays that cover the entire space of expression and used them to control the concentrations of GFP and RFP expressed from the chromosome (Fig 7A). As expected, the repression of one gene is independent of the repression of the other (Fig 7B). Strong correlations between GFP and RFP in single cells targeted with the same combinations of guide RNAs are due to common sources of extrinsic noise (Elowitz *et al*, 2002). We anticipate this to be a useful tool to study interactions of genes and specifically the effect of stoichiometry in genetic networks.

# Discussion

Here, we demonstrate that tuning gene expression through complementarity between guide RNA and target works robustly at the single-cell level, with two specific advantages over previous methods: First, relative repression strength is independent of native expression levels, making the system applicable to study genes of vastly different promoter strengths. Second, the system preserves endogenous expression noise of the repressed gene. This allows studying the impact of gene repression on cellular physiology without generating stochastic cell-to-cell variability, which is known to have important downstream consequences for processes such as cell differentiation or the emergence of spatial structure in populations (Çağatay *et al*, 2009; Waite *et al*, 2016).

The ability to control gene expression level through guide RNA complementarity rather than the concentration of an inducer has other advantages: For example, it enables differential control of different cells within the same culture. This could prove useful in pooled screens or competition assays. Furthermore, the strategy can be multiplexed to enable the simultaneous control of multiple genes independently without requiring multiple chemical inducers. We demonstrated this ability with two targets (GFP and RFP; Fig 7), which can be inserted in front of genes of interest. The strategy can easily be extended to include more than two fluorescent reporters as targets, as demonstrated in Appendix Fig S4.

Alternatively to using fluorescent-protein fusions it is also possible to guide dCas9 directly to the gene of interest, but this comes with the disadvantage of uncertainty about the exact repression strength due to two reasons: first, the rate at which dCas9 blocks the RNAP is dependent on the specific target sequence. In the future, it might thus be desirable to develop computational means

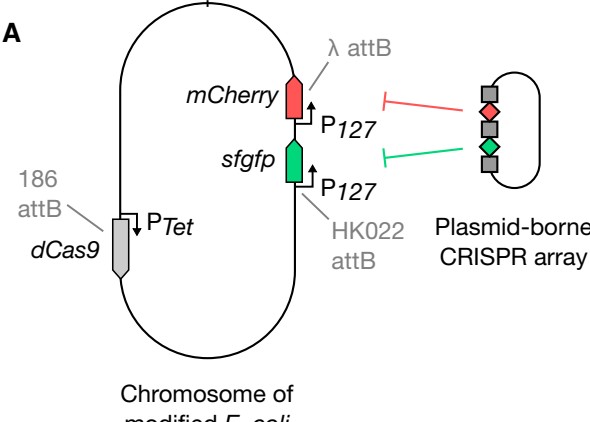

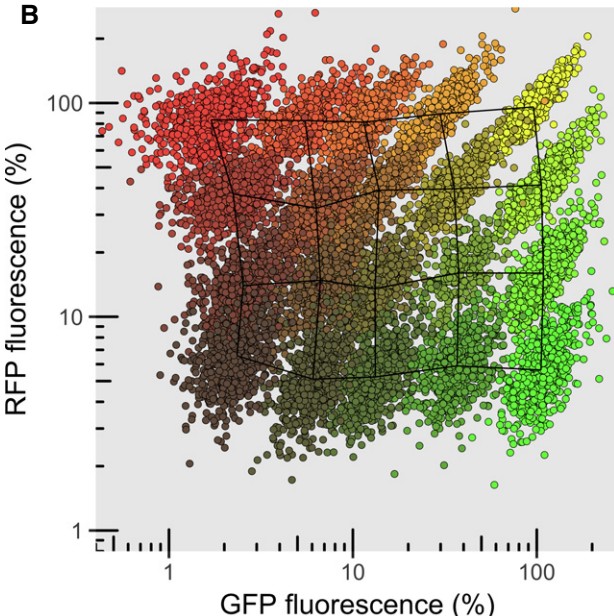

**Figure 7. CRISPR knockdown can be multiplexed to modulate expression of two genes without cross-talk.**

A   Schematic of the strain expressing two reporters and P_Tet-*dCas9* integrated in the chromosome at phage attachment sites. The levels of the two reporters can be controlled using a plasmid-borne CRISPR array coding for guide RNAs (diamonds) interspaced with CRISPR repeat motifs (squares), and also carrying the tracrRNA sequence (not shown).

B   Relative GFP and RFP concentration given relatively to the non-targeting spacer measured by high-throughput microscopy for a collection of 20 CRISPR plasmids. Each point represents a single cell, and each color represents the population obtained with one CRISPR plasmid. The overlaid meshwork connects the median values of the different populations.

Source data are available online for this figure.

to predict target repression based on sequence alone (Boyle *et al*, 2017). Second, any feedback controlling the expression of the target could lead to altered transcription-initiation rates (Fig 5). Therefore, using fluorescent-protein fusions has the advantage to report the exact expression level.

The properties of dCas9 repression described in this study originate from the mechanism of dCas9 binding to DNA inside ORFs. We

show that at high concentrations, dCas9 is saturating the target site even when using guide RNAs with large numbers of mismatches, where repression of the target gene is weak. We explain these observations by a "kick-out" model of repression, according to which RNAP kicks out dCas9 with a probability that can be tuned by spacer complementarity. The exact passage probability depends on the crRNA sequence. Here we provide a collection of guide RNAs against *mcherry* and *sfgfp* with known passage probabilities. Further work and larger datasets of diverse sequences will help to design new guides with predictable repression strength on arbitrary targets. The model predicts that repression is promoter-strength dependent in non-saturating conditions, in quantitative agreement with experiments (Fig 3C).

For full complementarity, our model is compatible with the hypothesis that dCas9 never leaves the target spontaneously but gets kicked out either by the RNAP or during DNA replication, consistent with the long half-life of dCas9 binding recently observed by single-molecule tracking and by restriction-protection assays (Jones *et al*, 2017) and previously also observed *in vitro* (Sternberg *et al*, 2014). A recent high-throughput study of dCas9 off-target binding and unbinding suggests that mutations in the PAM-distal region control the unbinding kinetics of dCas9 (Boyle *et al*, 2017). Since unbinding is dominated by RNAP-dCas9 collisions for the promoters tested here and as rebinding to the target is fast at high dCas9 concentrations, spontaneous unbinding plays no significant role for gene repression in our model system. However, if dCas9 targets the promoter rather than the coding region, spontaneous unbinding or replication-fork-based displacements might be the only modes allowing residual gene expression. This view is supported by the higher repression strength observed when targeting the promoter region (Bikard *et al*, 2013; Qi *et al*, 2013) and by the dependence on concentration (Appendix Fig S5).

Our results also suggest that dCas9 ejection does not lead to bursts of transcription, but that instead dCas9 returns to the target site after ejection in a time that is small with respect to the typical time interval between transcription initiations. It is still conceivable that such bursts may occur for transcription rates higher than the strongest promoter we used.

The results presented here argue for the use of high levels of dCas9 when performing CRISPRi assays in order to ensure that the target position is saturated. It is however important to highlight that the overexpression of dCas9 has been reported to be toxic for *E. coli* (Nielsen & Voigt, 2014). It is therefore preferable not to overexpress dCas9 far above the saturation point.

Our strategy enables to precisely control gene expression without introducing cell-to-cell variability, and should be useful for any quantitative measurements that depend on the expression level of a gene. By taking advantage of the ability to precisely control the rate at which dCas9 blocks the RNAP we could characterize a synthetic feedback loop, revealing unexpected properties of PhlF repression activity. In a second example, we took advantage of the ability to fine-tune expression levels at the steady state to quantitatively measure cell shape as a function of the levels of PBP2 and RodA. The level of precision achieved here would be hard to establish with conventional methods. Accordingly, this is the first study to establish a quantitative relationship between the abundance of cell-wall synthesizing proteins

and cell morphology at the population and single-cell levels. We anticipate that our method will be useful to study many other systems and in particular genetic circuits that include high levels of noise, such as stochastic switches, or other noise-dependent processes, where preservation of a well-defined level of expression noise is desirable.

# Materials and Methods

### Kick-out model of CRISPR knockdown

As already described in the main text, dCas9 is thought to bind to target DNA where it provides a roadblock for RNAP, thus blocking transcription. According to this mechanism, the expression level $\gamma$ of a dCas9-targeted gene is given by

$$\gamma = \gamma_0[1 - P(\text{stop})P(\text{bound})]. \tag{1}$$

Here, $\gamma_0$ is the native transcription rate, $P(\text{stop})$ is the probability of dCas9 blocking RNAP if dCas9 is occupying the target, and $P(\text{bound})$ is the probability that dCas9 is occupying the target (henceforth also termed occupancy).

While the probability $P(\text{stop})$ only depends on guide RNA-target complementarity, the occupancy $P(\text{bound})$ generally depends on complementarity, dCas9 concentration, and possibly on transcription-initiation rate (see the following paragraph). Therefore, repression is independent of dCas9 concentration only if the occupancy is very close to one ($1 - P(\text{bound}) \ll 1$), that is, if the target is saturated.

According to straight-forward reaction kinetics, the occupancy is given by

$$P(\text{bound}) = \frac{k_{\text{on}}[\text{dCas9}]}{k_{\text{on}}[\text{dCas9}] + k_{\text{out}}}, \tag{2}$$

where $k_{\text{out}}$ is the rate of dCas9 leaving the target. dCas9 can in principle leave the target by two different mechanisms, by transcription-independent unbinding (with rate $k_{\text{off}}$ if bound to the target), or by being kicked out from the target during all or part of the successful RNAP passage events, that is, during collisions where transcription continues. Alternatively, dCas9 could stay bound during all successful passage events. According to the two different models, $k_{\text{out}}$ is given by

$$k_{\text{out}} = \delta[1 - P(\text{stop})]\gamma_0 + k_{\text{off}}, \tag{3}$$

where we introduced an ejection frequency $\delta$. If $\delta = 1$, all successful passage events lead to dCas9 ejection. On the contrary, $\delta = 0$ corresponds to the scenario, where dCas9 stays bound during successful passage events. If the kick-out model was correct ($\delta > 0$), higher transcription-initiation rates $\gamma_0$ should thus lead to lower target occupancy. If the spontaneous unbinding model was correct ($\delta = 0$), occupancy should be independent of $\gamma_0$.

To identify the correct collision mechanism, we measured repression of *msfgfp* placed under two promoters with different promoter strengths at an intermediate level of dCas9 concentration, where the

target is not saturated by dCas9 and where changes of occupancy due to promoter strength should be clearly visible. We found that the relative GFP expression (normalized with respect to the unrepressed case) of a promoter with 2.6-fold higher promoter strength is increased by 1.7-fold as compared to the weaker promoter for full complementarity between guide RNA and target (Fig 3C). Thus, repression shows a strong dependence on promoter strength. This observation suggests that dCas9 is kicked out of the target site during all or part of the successful passage events. Plugging equation (2) into equation (1) and using equation (3) for $k_{out}$, the kick-out model of dCas9-based gene repression thus predicts a normalized transcription rate

$$\gamma^* = \frac{1 - P(\text{stop})k_{on}[\text{dCas9}]}{k_{on}[\text{dCas9}] + \delta[1 - P(\text{stop})]\gamma_0 + k_{off}}, \tag{4}$$

where we defined $\gamma^* = \gamma/\gamma_0$. Equation (4) is the central result of our model.

*Quantitative comparison with the experiment*
Here, we compare model prediction and experimental expression rates for two different promoters in saturating and non-saturating conditions (Fig 3B and C, respectively). We first consider the case of full complementarity between guide RNA and target. For a quantitative comparison, we eliminate one of the three experimentally unknown parameters, the dCas9 rebinding rate $k_{on}[\text{dCas9}]$, by introducing the following dimensionless quantities: $\lambda = \delta\gamma_0/(k_{on}[\text{dCas9}])$ is the ratio of the rate of induced dCas9 displacements over the rate of dCas9 rebinding, $\alpha = k_{off}/(k_{on}[\text{dCas9}])$ is the ratio of transcription-independent unbinding rate over rebinding rate, and $r = 1 - P$ (stop) is the probability of successful transcription in the presence of target-bound dCas9. The latter probability is known to be $r = 0.026 \pm 0.003$ for full complementarity between guide RNA and target from independent experiments in saturating conditions (Fig 3B). Equation (4) can then be written as

$$\gamma^* = \frac{r + \lambda r + \alpha}{1 + \lambda r + \alpha} \tag{5}$$

leaving two unknown parameters, $\lambda$ and $\alpha$.

The lifetime of dCas9-DNA complexes is greater than 45 min *in vitro* (Sternberg *et al*, 2014), suggesting that the unbinding rate $k_{off}$ is low *in vivo*. We thus hypothesized that the dimensionless transcription-independent unbinding rate might be negligibly small, that is, $\alpha \ll 1 + \lambda r$, which would allow us to simplify

$$\gamma^* \approx \frac{r + \lambda r}{1 + \lambda r}, \text{ if } \alpha \ll 1 + \lambda r \tag{6}$$

thus leaving only one unknown parameter $\lambda$.

To test this hypothesis, we compare the predicted expression level (equation 5) for zero and finite values of $\alpha$ to our experimentally obtained data of GFP expression from the two promoters $P_{PhlF}$ and $P_{127}$ (Fig 3C). In Appendix Fig S7A, the predicted expression $\gamma^*$ is plotted as a function of $\lambda r$ for two values of $\alpha$ ($\alpha = 0$, $\alpha = 0.3$).

To obtain the prediction of GFP expression, we used the measurements of $P_{127}$-GFP as a reference to infer the normalized

kick-out rate $\lambda(P_{127})r$ corresponding to the experimental GFP expression levels of $\gamma^*_{ex}(P_{127}) = 0.32 \pm 0.04$ (Fig 2C),

$$\lambda(P_{127})r = \frac{(1 + \alpha)\gamma^*_{ex}(P_{127}) - r - \alpha}{1 - \gamma^*_{ex}(P_{127})} \tag{7}$$

also indicated by the gray vertical lines in Appendix Fig S5A for the two values of $\alpha = 0$ and $\alpha = 0.3$. Then, we asked for the predicted relative expression level of the repressed $P_{PhlF}$-GFP, given prior knowledge that the $P_{PhlF}$ promoter is $(2.6 \pm 0.2)$-times stronger than $P_{127}$ (Fig 3A) and therefore $\lambda(P_{PhlF}) = (2.6 \pm 0.2)\lambda$ $(P_{127})$. The predicted expression level of the $P_{PhlF}$ promoter is thus

$$\gamma^*(P_{PhlF}) = \frac{r + \lambda(P_{PhlF})r + \alpha}{1 + \lambda(P_{PhlF})r + \alpha}. \tag{8}$$

According to our hypothesis of $\alpha = 0$, we obtain $\gamma^*(P_{Phlf}) = 0.54 \pm 0.05$. This value is in great quantitative agreement with the measured expression level of $\gamma^*_{ex}(P_{PhlF}) = 0.54 \pm 0.02$. On the contrary, values of $\alpha > 0.14$ lead to predicted expression levels significantly lower than the experimental value (Appendix Fig S7B). Together, these results confirm that the kick-out model quantitatively describes the mechanism of dCas9 repression and suggest that the transcription-independent unbinding rate for guide RNAs of full complementarity to their targets is indeed much lower than the rebinding rate ($k_{off} \ll k_{on}[\text{dCas9}]$).

We then wondered whether the transcription-independent unbinding rate $\alpha$ would increase for reduced degrees of complementarity. To that end, we performed the same analysis as above on experimental data for GFP expression from the two promoters at non-saturating levels of dCas9 but now for guide RNA that carries six mismatches (corresponding to a increased passage probability of $r = 0.056 \pm 0.001$). As in the case of full complementarity, we found an excellent agreement between the experimental and predicted GFP expression levels ($\gamma^*_{ex}(P_{PhlF}) = 0.79 \pm 0.04$ vs. $\gamma^*$ $(P_{PhlF}) = 0.77 \pm 0.03$) for a transcription-independent unbinding rate of $\alpha = 0$ while predicted expression values for $\alpha > 0.35$ are significantly lower than the measured expression level. We also carried out experiments with a guide RNA of 10 mismatches. However, the experimental data showed uncertainties too large to make conclusions about the level of $\alpha$. We thus conclude that the dCas9-target complex is stable even for reduced degrees of complementarity of down to 14 bp and maybe less.

*A lower limit for the lifetime of the dCas9-DNA complex*
To obtain a quantitative estimate of the rate $k_{off}$ in the case of full complementarity, we took advantage of the relationship

$$k_{off} \lesssim 0.14 k_{on}[\text{dCas9}] = \frac{0.14\delta\gamma_0}{\lambda} \approx 8 \times 10^{-3}\delta\gamma_0,$$

where we used $\lambda r(P_{127}) \approx 0.43$ and $r \approx 0.026$. $P_{127}$ is based on the consensus promoter sequence and thus not expected to be stronger than well-studied model promoters in *E. coli*, which display transcription rates of not more than about $20 \text{ min}^{-1}$ (Kennell & Riezman, 1977; So *et al*, 2011). This suggests that the unbinding rate of dCas9 in the absence of RNAP-based collisions is smaller than $1/6 \text{ min}^{-1}$. This is based on the conservative assumption that

all successful transcription events lead to a displacement of dCas9 from the target, that is, $\delta = 1$. For the reduced complementarity of 6 mismatches we obtain a lower limit of $1/2.4$ min$^{-1}$.

The average number of chromosomal loci per cell coding for *msfgfp* is about 2, given a doubling time of about $t_d \approx 30$ min in our growth conditions, an average DNA replication time of about $t_C \approx 60$ min at 30°C (Breier *et al*, 2005), and assuming a D-period of $t_D = 20$ min. The gene copy number per cell is then given by $n(\Phi) = 2^{t_C(1-|\Phi|)} + t_D \approx 2$, where $\Phi = 0.66$ is the relative distance of the gene locus from the origin of replication, if distance is normalized with respect to the distance between origin and terminus. Thus, the lifetime of a single dCas9-DNA complex ($n/k_{off}$) is greater than about 12 min (full complementarity) or 5 min (6 mismatches). Notably, these timescales should be regarded as lower bounds, while the actual lifetimes might be significantly larger. This estimate is thus in agreement with recent single-molecule tracking and restriction-protection assays (Jones *et al*, 2017), which demonstrate that the lifetime of the dCas9-DNA complex is equal to the cellular doubling time in the case of full complementarity. It is also in agreement with *in vitro* data, which suggest that the lifetime is greater than 45 min (Sternberg *et al*, 2014). Given that the lower limit of the *in vivo* lifetime is of the same order of magnitude as the cell doubling time of 30 min, it is conceivable that dCas9 virtually never leaves the target site by equilibrium unbinding but instead is kicked out during DNA replication events by the DNA replication machinery, as also strongly supported by (Jones *et al*, 2017).

### Genome modifications

All the strains used for measurements derive from *E. coli* MG1655. Appendix Table S1 details the construction of the strains used in this study.

For integration of cassettes at phage attachment sites, we used the "clonetegration" method (St-Pierre *et al*, 2013). Integrated backbones were excised by expressing a flippase from pE-FLP. Plasmid pLC97 can be used to easily integrate P$_{tet}$-*dCas9* at the lambda attB site.

For scarless integration of *mCherry-mrdA* in the native *mrdAB* operon, we used the pCas/pTarget system (Jiang *et al*, 2015). The PAmCherry-PBP2 protein fusion present in strain TKL130 (Subach *et al*, 2009; Lee *et al*, 2014) was replaced by a translational fusion with *mCherry* extracted from plasmid pFB262 (Bendezú *et al*, 2009). To this end, the pAV06 variant of pTarget was constructed and genome editing was performed as described in reference (Jiang *et al*, 2015). The deletion of the *lacY* gene in AV76 was done by P1 transduction using the strain JW0334 from the Keio collection (Baba *et al*, 2006) as a donor.

We also propose a novel CRISPR-Cas9 allelic exchange strategy for the scarless integration of *mCherry* or *sfgfp* in front of genes of interest. This strategy is detailed in the Appendix Fig S9.

The sequences of the *sfgfp* and *mCherry* genes used in this study can be found on the GenBank database with accession codes KT192141.2 and JX155246.1, respectively.

### Plasmid design and construction

The CRISPR targets were chosen next to the beginning of the ORF, but at least 50 bp away from the initiation codon, in order to

preclude unwanted interactions with the native regulation of transcription. None of the spacers used in this study have any off-target position with more than 8 bp of complementarity in the PAM-proximal region. Spacers were cloned into the CRISPR array of plasmid pCRRNA using Golden-Gate assembly as previously described (Bikard *et al*, 2014; Cui & Bikard, 2016). The oligonucleotide sequences are available in Appendix Table S4.

The other plasmids from this study were constructed by Gibson assembly (Gibson *et al*, 2009). The fragments are described in Appendix Table S2 and the primer sequences in Appendix Table S3. DNA constructions were electroporated in *E. coli* strain DH5α or Pi1 for *pir*-dependent origins of replication (Shafferman & Helinski, 1983).

### Media and reagents

For all flow cytometry measurements, the cells were grown in Luria-Bertani (LB) broth. As a minimal medium for the *mrdAB* measurements, we used M63 medium supplemented with 2 g/l of glucose, 10 mg/l of thiamine, 10 mM of MgSO$_4$, and 1 g/l of casaminoacids. When needed, we used various antibiotics (25 μg/ml chloramphenicol, 100 μg/ml carbenicillin, 50 μg/ml kanamycin, 100 μg/ml spectinomycin). Di-acetyl-phloroglucinol (DAPG), anhydrotetracycline (aTc), and isopropyl β-D-1-thiogalactopyranoside (IPTG) were used, respectively, for induction of P$_{PhlF,}$ P$_{Tet}$, and P$_{Lac}$ promoters. All oligonucleotides were obtained from Eurofins Genomics.

### Preparation of steady-state exponential cultures

Unless stated otherwise, all cultures were grown at 30°C. Strains were first re-streaked from a freezer stock. Independent single colonies were picked for each replicate. Cells were then grown overnight in 96-deep-well plates using a tabletop shaker in 1 ml of medium with 100 ng/ml of aTc and 50 μg/ml of kanamycin (Eppendorf). The day of the measurement, cultures were back-diluted 250 times in fresh medium with aTc and kanamycin, and grown for 1 h 45 min into exponential phase. We then fixed the cells with 4% formaldehyde (30 min on ice) and washed with phosphate-buffered saline (PBS).

### Growth rate measurements

To determine the doubling times of *E. coli* with various induction levels of dCas9, we prepared the cells into steady-state exponential growth then diluted the cultures 1/250 in a flat-bottomed 96-microwell plate (Greiner) and recorded optical density along growth using a microplate reader (Tecan). We fitted an exponential function to the data points corresponding to the exponential phase in order to calculate the doubling time.

### Flow cytometry

Fluorescence of single cells was recorded using a Miltenyi MACS-quant flux cytometer. 10,000 events were recorded per replicate. In all cases, the AV01 strain (with no reporters) carrying a non-targeting pCRRNA plasmid was used to measure the auto-fluorescence background. We calculated the mean fluorescence signal of each population and subtracted the mean auto-fluorescence signal.

To test whether differences in expression were significant, we performed Student's *t*-test on the natural logarithms of the average fluorescence (Beal *et al*, 2016).

## High-throughput microscopy (imaging cytometry)

An Amnis ImageStreamX (EMD Millipore) imaging cytometer was used to image the cells in high-throughput in brightfield, GFP, and RFP channels. Images were analyzed using the IDEAS® (EMD Millipore) software suite. For each condition, at least 10,000 events were recorded per replicate. Cells that were out of focus or tilted were identified by calculating the average gradient of a pixel normalized for variations in intensity levels (*Gradient RMS* feature in IDEAS®). Additionally, we used the Feature Finder script of IDEAS® to remove contaminating particles, images with multiple cells and beads. After filtering, at least 2,000 images remained per sample. The fluorescence channels were not used for filtering. A color compensation matrix was calculated to account for spectral overlap of GFP and RFP emission spectra, so cultures of AV02 (GFP only) and AV04 (RFP only) would each have a null signal on the converse channel. As a proxy for the reporter's intracellular concentration, we used the average image intensity inside the area corresponding to each cell. The cell area was determined by using a threshold on the bright field images. Single points located more than three standard deviations away from the population average were discarded as outliers, as they can disrupt the noise computations. The average fluorescence $\mu$ of each sample was calculated by taking the mean of the single-cell fluorescence. The noise was defined as $\sigma/\mu$, with $\sigma = \sqrt{\sigma_{sample} - \sigma_{blank}}$ where $\sigma_{sample}$ is the standard deviation of the intracellular average intensity of the sample, and $\sigma_{blank}$ is the standard deviation of a sample with no fluorescent reporter (noise from auto-fluorescence).

## Fluorescence and phase-contrast microscopy

Fixed cells were transferred to PBS microscopy pads with 1.5% UltraPure Agarose (Invitrogen) and imaged using an inverted microscope (TI-E, Nikon Inc.) equipped with a 100× phase-contrast objective (CFI PlanApo LambdaDM100× 1.4NA, Nikon Inc.), a solid-state light source (Spectra X, Lumencor Inc.), a multiband dichroic (69002bs, Chroma Technology Corp.). mCherry fluorescence was measured using excitation (560/32) and emission (632/60) filters. Images were acquired using a sCMOS camera (Orca Flash 4.0, Hamamatsu) with an effective pixel size of 65 nm.

MATLAB code adapted from the Morphometrics package (Ursell *et al*, 2017) was used to find cell contours from phase-contrast images. Background intensity, uneven illumination, and cell auto-fluorescence were accounted for in the analysis. For the analysis of fluorescence signal, we corrected the raw mCherry values for uneven illumination, background intensity, and cell auto-fluorescence. Intracellular protein concentration was obtained as the mean pixel intensity inside the cell area. Total regression was used to find the major axis of the cell. Cell width was defined as the average distance between the cell contour and this axis, excluding the poles.

Measurements of cell morphology during steady-state exponential growth (Appendix Fig S12) were performed after overnight induction of dCas9, followed by 1/250 dilution of the culture in fresh M63 medium with aTc and while the culture was kept in exponential growth phase at an optical density below 0.1. Samples were taken from the culture, fixed, and imaged after 2 and 4 h.

## Northern blot

Total RNA was extracted from cultures in early stationary phase using TRIzol. Electrophoresis on Novex® TBE-Urea Gels (10% polyacrylamide gels containing 7 M urea, Invitrogen) was used to separate RNAs. The gels were blotted onto Nylon membranes (Invitrogen), which were subsequently cross-linked with 1-ethyl-3-(3-dimethylaminopropyl) carbodiimide (EDC, Thermo Scientific) buffer (Pall & Hamilton, 2008). The probes were labeled as follows: 100 pmol of oligonucleotide was heat denatured, labeled, and phosphorylated by mixing 40 $\mu$Ci of $^{32}$P-$\gamma$-ATP (PerkinElmer) and T4 PNK (NEB) reagents. A labeled probe specific to the guide RNA R20 (5′ GCATAGCTCTAAAACTCCGTATGAAGGCACCCAGA 3′) was column purified (Macherey-Nagel PCR cleanup kit) and used for overnight hybridization. The intensity of the shortest band, corresponding to the fully processed guide RNA, was quantified using the Fiji software package.

**Expanded View** for this article is available online.

## Acknowledgements

We thank T. K. Lee and K.C. Huang for providing strain TKL130, F. Bendezù for providing plasmid pFB262, H. Cho and T. Bernhardt for providing plasmid pHC942, and D. Mazel for providing plasmid pSW23t. We thank J. Fernández-Rodríguez for providing the P$_{PhlF}$–*sfgfp-phlF* DNA fragment. We thank E. Brambilla and E. Oldewurtel for support in microscopy, as well as A. Soler and M. Hasan for support regarding flux cytometry and imaging cytometry. We acknowledge the Technology Core of the Center for Translational Science (CRT) at Institut Pasteur for support in conducting this study. This work was supported by the European Research Council (ERC) under the Europe Union's Horizon 2020 research and innovation program [Grant Agreement No. (677823) to DB and No. (679980) to SvT]; the French Government's Investissement d'Avenir program Laboratoire d'Excellence "Integrative Biology of Emerging Infectious Diseases" (ANR-10-LABX-62-IBEID) to SvT, DB, and AV; the Marie de Paris "Emergence(s)" program and the Volkswagen Foundation to SvT; and the Pasteur-Weizmann consortium to DB.

## Author contributions

DB, SvT, and AV designed the study. AV constructed the strains and plasmids and performed all measurements and data analysis. SvT developed the mathematical models. LC took part in strain construction, and EO performed the compensation of uneven illumination for the measurement of mCherry-PBP2 by microscopy. AV, DB, and SvT wrote the manuscript.

## Conflict of interest

The authors declare that they have no conflict of interest.

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
