## [Review Process File · Molecular Systems Biology]

Tuning dCas9's ability to block transcription enables robust, noiseless knockdown of bacterial genes

Antoine Vigouroux, Enno Oldewurtel, Lun Cui, David Bikard and Sven van Teeffelen

Review timeline:

Submission date:	4 August 2017
Editorial Decision:	25 September 2017
Revision received:	31 January 2018
Editorial Decision:	7 February 2018
Revision received:	8 February 2018
Accepted:	14 February 2018

Editor: Maria Polychronidou

Transaction Report:

1st Editorial Decision

25 September 2017

Thank you again for submitting your work to Molecular Systems Biology. We have now heard back from the three referees who agreed to evaluate your study. As you will see below, the reviewers appreciate that the presented findings are relevant for the scientific community. They raise however a series of concerns, which we would ask you to address in a revision of the manuscript.

The reviewers' recommendations are rather clear so I think that there is no need to repeat all the points listed below. Reviewer #1 recommends performing biophysical analyses in order to more convincingly support the mechanism involving RNAP/dCas9 collision and dCas9 displacement. However, we think that while these experiments would enhance the impact of the study, they are not mandatory for the acceptance of this work. Of course, we would not be opposed to the inclusion of such data, in case you wish to perform experiments in the direction suggested by reviewer #1. Please do not hesitate to contact me if you need to further discuss any of the issues raised by the reviewers.

REVIEWER REPORTS

Reviewer #1:

dCas9 has recently emerged as a critical tool in genetic engineering. The manuscript of Vigouroux et al performs some interesting modeling and experimental analysis to better understand the mode of repression and interaction. There are some good insights in this paper and the data is actually quite nice. At a high level, the topic area, data, and model are appropriate for MSB. The core problem is that the authors overstate the detailed mechanism of dCAS9 repression, which is not fully supported by the data. It is noted that the senior author has not previously published in this space and may not

fully appreciate the experiments needed to validate the (interesting) hypotheses put forward. There are two routes that the authors could take: 1. Reduce the claims of the paper and focus on what can be drawn from the performed experiments (a deeper analysis of the feedback loop, cell wall control, and altering repression via guide RNA mismatches are all interesting routes), or 2. Perform the significant biophysical analysis required for fully support their current claims.

The key concern is around the claims regarding the collision and displacement of dCas9 bound to a promoter by RNAP. This is very interesting and could be a significant result. However, it can not be inferred from the data and the model presented. While this may provide initial insight, it needs to be confirmed with follow-up biophysical work. The transcription field, notably with terminators, has extensive work in this area.

Additional comments:

1, In Figure 1c, it is odd why higher expression of dCas9 showed narrower population distribution whereas lower expression resulted in wider distribution. The authors need to explain the difference.

2, The authors claimed that using complementary-based guide RNA could reduce noise levels, while varying dCas9 concentrations generated more noise. However, in Figure 2b, the data points are quite noisy compared to Figure 1a, which is inconsistent with the authors' claim that using complementary base guide RNA generates less noise.

3, The authors developed a model for the complementary sgRNA based strategy and the model worked well for spacers that have 14 or 20 complementary base pairs. However, the model doesn't fit the results with 10 complementary base pairs, even though the repression abilities were consistent among samples as shown in Figure 3b (and Figure 1a). Is this model limited to high complementarity cases? In order for this model to be useful, the author should include the data for 10 complementary base pairs (possibly also 11 base pairs), and clarify why the model didn't work. Are there some other limitations in this model?

4, The authors also came up with a model for the feedback loops. Interestingly, the model suggested that increased DAPG concentration strangely stabilizes dimerization of PhIF for repression. This needs to be verified by experiments to validate the model.

5, In Figure 5, the author used cooperativity $n=1$ or $n=2$ for PhIF to fit the data in the feedback loops. However, $n=4.5$ was observed previously for PhIF. The authors need to address this discrepancy. In addition, it is not clear why $a=0.15$ was used in Figure EV6 to fit the model.

6. The interpretation of the Elowitz 2002 paper is often stretched to the point of being wrong. The comment regarding the expression noise difference between inducible and native promoters is not generally correct and this reference only looks at a few examples. The citation on lines 293-294 is also incorrect, but could be due to a grammatical problem or misplacement of the reference.

7. dCas9 can be very toxic when over expressed in the host. The ability to integrate it into the genome under inducible control is interesting and would be very useful. To what extent is there a growth impact at different levels of aTc induction? How stable is the strain when being passaged over multiple generations?

Reviewer #2:

Vigouroux et al. reveal the underlying importance of the complementarity between the crRNA of CRISPR/Cas9 system to the target sequence for noiseless targeted gene repression. Their approach also leads to a new mechanistic model that accurately describes dCas9-mediated repression in bacteria. Importantly, the authors found that it is the extent of mismatch between the crRNA and the target gene that controls the degree of repression of the target gene independent of promoter strength, enabling fine-tuned controlled of (presumably) any gene expression. The authors also demonstrate that a "kick-out" model of dCas9 binding that does not rely on dCas9 concentration, but rather the degree of complementarity to the target sequence. This is an excellent study and I am

supportive of publication assuming the following concerns are addressed.

1. It would be very useful to contrast these dCas9 results with alternative transcription factor-based control of the same gene. Specifically, can the authors reproduce Fig 1a-d using TALEs for repression and mutations in the target? Alternatively, LacI-LacO system with the known mutations in the LacO sequence that reduce the affinity.
2. Figure 5. Does not depict the various mismatch guides used, only the varying amount of DAPG.
3. Targeting the *mrdaB* operon for repression does not reduce copy number if the construct is integrated, i.e. not on a plasmid.
4. Figure 1a. What is the concentration of aTc for the various guides?
5. Figure 2. Decoy guides. How many active guides are in the cell? They integrated the array, with at least 2 guides for GFP and RFP, but only added one array. Important point because the authors claim that this approach can be multiplexed

Reviewer #3:

the conclusions to be accepted as stated (see concerns in next section). Review of "CRISPR-dCas9 enables noiseless, fine-tuned and multiplexed repression of bacterial genes"

Summary

This manuscript aims to show that endogenous genes can be repressed in microbial systems in a manner that does not increase transcriptional noise and is based on the degree of guide-RNA sequence complementarity, rather than repressor expression level. A consistent relative repression of a target gene can be generated by high (saturating) levels of dCas9 binding and blocking RNA polymerase. Repression strength is modulating by creating mismatches in the guide-RNA -- target DNA sequence that increases the probability that dCas9 will unbind when encountering a transcribing RNA polymerase (the "kick-out" model). Further, temperature was shown to affect the kick-out rate. An established auto-regulatory feedback loop (PhIF) was used to demonstrate how repression under the kick-out model is dependent on transcription initiation rate. Additionally, this gene repression paradigm was used to knock-down two genes involved in cell shape and was shown to be able to modulate two promoters independently using guide-RNAs with different numbers of mismatches. Technically, the experiments were almost exclusively based on live, high-throughput imaging of the expression of fluorescent reporter proteins in populations of microbes.

General remarks

The data, as far as it goes, is convincing and the experiments are elegant and well-designed. I would recommend this manuscript for publication, however, some key controls and variables need to be included for

CRISPR-Cas9 systems are becoming an important research tool for dissecting genetic mechanisms in biology. This manuscript demonstrates how inactive Cas9 systems can be used to repress genes with high controllability. The advances in this manuscript are both conceptual and technical in nature. The "kick-out" model of dCas9 unbinding is mechanistically interesting in that suggests that differing guide-RNA mismatches is more controllable as to repress genes than changing repressor concentration. The technical advance is demonstrating the controllability of gene repression: consistent, relative fold-reductions in gene expression and a lack of increase in transcriptional noise introduced by the repression mechanism. This difference in transcriptional noise between repression based on guide-RNA mismatches and repressor concentration was a striking result and will be an important finding for the broader research community.

The advance in fine-scale controllability of gene repression is a significant advance compared to more established methods of repression by increasing repressor concentration. This work will likely be implemented by laboratories interested in gene repression in microbial systems. All audiences interested in modulating gene expression levels in microbial systems would be highly interested in this work, including and especially synthetic biologists constructing gene circuits. Research in eukaryotes will also benefit from considering the data in this manuscript, although the situation is more complex in eukaryotes.

Major concerns

- A central conclusion of this manuscript is that at saturating conditions, the only determinant of relative repression is the degree of complementarity between guide-RNA and DNA (i.e. - the kick-out model). Figure 2 is purported to demonstrate this, however more extensive controls are needed for such a critical experiment. Specifically, in Figure 2a, a dummy guide is used to reduce active dCas9 levels, but only indirect evidence (the amount of processed guide-RNA) is provided and only for 2 conditions (a two-fold difference in active dCas9 level). Crucially, the amount of active-dCas9 should be directly measured through a dCas9-pull-down coupled to northern blot, quantitative-PCR, or sequencing. This can show that the targeting sequence and control sequence each comprise half of the dCas9-RNPs in the cell. Additionally, a more convincing demonstration of relative repression independence would include higher and lower levels of active dCas9 (until the point of sub-saturating conditions). This point impacts the interpretation of Figure EV3 as well (whether temperature is modulating repression strength via differential occupancy or P(stop)).

- The effect of sequence content variability within the gRNA-target region should be addressed. This manuscript describes the effect of the number of mismatches, but not whether the identity of mismatching base-pairs affects the probability that dCas9 will stop transcription, P(stop). For example, does an eliminated G-C base pairing decrease P(stop) more than an eliminated A-T base-pairing in the complementary region due to its higher thermodynamic binding energy? This adds another dimension to the relative repression model and affects the generalizability of this paradigm for repressing arbitrary genes.

- Figure 3 and EV2: Do these results extend to different promoters with different transcription rates? Only two have been tested and these differ in basal transcription rate by a factor of approximately 2-2.5. If these results are supposed to be generalizable to modulate the relative expression of arbitrary genes, a larger range of promoter strengths/transcription initiation rates should be tested as in Figure 3 (e.g. - 2-4 more promoters with higher and lower transcription rates).

Minor points

- Related to Figure 1: A semantic issue, but possibly misleading to readers who do not examine the raw data. The authors claim a "noise-less" repression is can be instantiated and controlled by the degree of complementarity between guide-RNA and target DNA. The data show a striking degree of noise introduced by repression based on dCas9 level, and that this extra noise is not generated by altering mismatch number, however the intrinsic biological noise of transcription is still present (as demonstrated by the overlapping distributions in Figure 1D). Therefore, it's more appropriate to refer to mismatch based repression as not adding additional noise to endogenous transcription rate, but the expression level is not noiseless.

- Figure 1f is not referenced or discussed in the text or supplement. This should be included.

- Figure 5: Missing labels. I'm guessing that the 5 data points on each line correspond to 20, 14, 11, 10, and 0-bp of complementarity (from left to right), but this must be included. Also, the figure says there is a "No Feedback" label on the graph (there is not), this corresponds to the "No PhIF" line, but the legend and figure should match.

- Line 145: Likely that figure 1d should be referenced instead of Figure 1c.

- Supplement: section "Quantitative comparison with experiment": Figure 2c is referenced, but this should be figure 3c.

- Supplement: section "Kick-out model of CRISPR knock-down": It would be helpful to readers to explicitly derived Equation 2.4.

- Supplement: EV2: better figure labeled is needed to interpret this graph. Which are the predicted and experimental points? What is signified by the vertical lines?

- Lines 267-269, Supplement EV8b: Considering how small the reduction in cell length was at high repression, I'm skeptical of the interpretation that the cell is minimizing the area:volume ratio to compensate for low PBP2. However, this does not negate the strong effect/conclusion of the cell

morphology vignette of this manuscript.

- While not required for interpretation of the data in this manuscript, an interesting experiment would be to compare mismatches occurring at the 5'-end of the guide-RNA vs internal mismatches (that do not disrupt the 3'-end seed region). Does 6-bp of mismatch at the 5'-end cause a different P(stop) value than an internal 6-bp mismatch (a does it matter if the mismatches are adjacent, as one block, or interspersed throughout the target region)? A difference here would require a refinement of the conclusion that the number of mismatches determines relative repression level and would need to include mismatch location as a variable.

1st Revision - authors' response

31 January 2018

Reviewer #1

dCas9 has recently emerged as a critical tool in genetic engineering. The manuscript of Vigouroux et al performs some interesting modeling and experimental analysis to better understand the mode of repression and interaction. There are some good insights in this paper and the data is actually quite nice. At a high level, the topic area, data, and model are appropriate for MSB. The core problem is that the authors overstate the detailed mechanism of dCAS9 repression, which is not fully supported by the data. It is noted that the senior author has not previously published in this space and may not fully appreciate the experiments needed to validate the (interesting) hypotheses put forward. There are two routes that the authors could take: 1. Reduce the claims of the paper and focus on what can be drawn from the performed experiments (a deeper analysis of the feedback loop, cell wall control, and altering repression via guide RNA mismatches are all interesting routes), or 2. Perform the significant biophysical analysis required for fully support their current claims.

The key concern is around the claims regarding the collision and displacement of dCas9 bound to a promoter by RNAP. This is very interesting and could be a significant result. However, it can not be inferred from the data and the model presented. While this may provide initial insight, it needs to be confirmed with follow-up biophysical work. The transcription field, notably with terminators, has extensive work in this area.

We thank the reviewer for appreciation of our work and for his/her critical comments. We are now more explicit about the conclusions that can be drawn from our different experiments by changing section titles and introducing a new section for the experiments performed at non-saturating dCas9 expression. The three corresponding sections now read

- "RNAP can transcribe dCas9-bound targets in a complementarity-dependent manner" (line 114),
- "If dCas9 is saturating the target, relative repression is independent of target gene promoter strength" (line 169)
- "If dCas9 is not saturating the target, relative repression depends on promoter strength, supporting a 'kick-out' model of dCas9 ejection by RNAP" (line 186)

The content of the latter section has been rewritten substantially to reflect this distinction and to discuss the validity of the kick-out model. We have also weakened the strong conclusion of the kick-out model. Finally, we included a new experiment to support the kick-out model further (see below). Summarizing, we are convinced that our experiments and model provide evidence to conclude the following:

- a) At high dCas9 concentration and for targets inside the gene, repression is independent of dCas9-complex concentration. Therefore, dCas9 is saturating the target at high dCas9 concentration and the reduction in gene repression due to the presence of mismatches must be due to dCas9 allowing transcription even if it is bound during a transcription-initiation event. Therefore, an occupancy-based model is strictly ruled out based on our observation. This still leaves the question open, whether dCas9 is either physically displaced from the target or remains bound to the coding strand while leaving the template strand free for RNAP passage.
- b) At low, non-saturating dCas9 expression levels we find that repression depends on promoter strength. For full and intermediate levels of complementarity this finding quantitatively agrees with the kick-out model, which assumes that dCas9 is physically

displaced from the target upon successful passage events. The alternative equilibrium-unbinding model, that the dCas9 complex stays bound to the target and remains intact to block successful transcription events, is strictly ruled out. We note, however, that it is possible that RNAP-dCas9 collisions only lead to partial displacement, i.e., that dCas9 is only displaced during a fraction of RNAP-dCas9 collisions. We now included this possibility in the main text and in the mathematical model by introducing a displacement fraction δ in Eq. 3 of M&M (line 458). This additional variable does not affect the comparison of model and experiments.

To support the proposed kick-out model of gene repression we have now additionally constructed a strain where the target sequence is placed inside the promoter region. In this case we do not expect any kick-out events, as RNAP is thought to bind this region due to diffusion and capture rather than due to processive polymerase activity. Indeed, we observe that repression is much stronger for a target inside the promoter region than for the same target inside the gene for high and intermediate degrees of complementarity, consistent with the expectation that no kick-out event happens inside the promoter region (see the new Appendix Figure S5 also included below).

Furthermore, we find that inside the promoter region, repression changes as a function of dCas9-complex concentration even for high dCas9 concentrations and for high degrees of complementarity (demonstrated for 20 and 11 bp of complementarity in the figure below). We thus conclude that any residual gene expression depends on the rare periods of dCas9 not being bound to the target due to spontaneous or replication-induced unbinding and delayed rebinding. The duration of these periods determines the rate of transcription according to our model, and these durations depend on dCas9 concentration.

The new data is now included in the manuscript, (see line 164 and Appendix Figure S5): We write “Interestingly, when the same target is moved from the ORF to the promoter region, repression is increased and depends on concentration of active dCas9 complex (Appendix Fig S5)” The new figure is also shown here:

Appendix Figure S5.

Additional comments:

1, In Figure 1c, it is odd why higher expression of dCas9 showed narrower population distribution whereas lower expression resulted in wider distribution. The authors need to explain the difference.

We thank the referee for pointing out this observation. At both high and low levels of induction, noise is lower than at intermediate levels. We attribute the increased fluctuations in gene repression to increased fluctuations in dCas9 levels at intermediate inducer levels: First, at high aTc inducer levels the Tet repressor never binds the operator site, rendering the promoter for dCas9 expression essentially constitutive. Therefore, fluctuations of dCas9 expression are expected to be low. Furthermore, repression is independent of dCas9 expression levels for high dCas9 expression (Fig 2). At intermediate levels of dCas9, fluctuations in dCas9 levels are expected to be higher, because dCas9 expression now depends on both the fluctuating concentrations of inducer and repressor protein. We do not know, which of those two fluctuations are predominantly responsible, but it appears plausible to us that gene-expression noise is higher in this regime, as any fluctuations of repressor or cytoplasmic inducer concentration would lead to additional fluctuations of gene expression. In the absence of inducer, inducer fluctuations do not contribute to gene-expression variations and noise accordingly goes down. We have added an explanation to the manuscript (line 97).

2, The authors claimed that using complementary-based guide RNA could reduce noise levels, while varying dCas9 concentrations generated more noise. However, in Figure 2b, the data points are quite noisy compared to Figure 1a, which is inconsistent with the authors' claim that using complementary base guide RNA generates less noise.

Data in Figure 1 are obtained by high-throughput microscopy, while data presented in Figure 2b are obtained by flow cytometry. Flow cytometry has lower accuracy than high-throughput microscopy and thus shows higher degrees of variability between replicates. This is to distinguish from the noise on gene expression within one population of cells, which is not increased by complementarity-based repression but increased by dCas9-concentration-dependent repression, as shown in Fig 1 and Appendix Fig S2.

We have now explicitly mentioned the use of the different techniques in the text by writing: "We note that these and the following measurements of population averages are performed by flow cytometry and are thus generally noisier than the results obtained by high-throughput microscopy presented in Fig 1." (line 132). In the paragraph on PBP2 we write "Single-cell measurements were then performed by phase-contrast and epifluorescence microscopy." (line 318).

3, The authors developed a model for the complementary sgRNA based strategy and the model worked well for spacers that have 14 or 20 complementary base pairs. However, the model doesn't fit the results with 10 complementary base pairs, even though the repression abilities were consistent among samples as shown in Figure 3b (and Figure 1a). Is this model limited to high complementarity cases? In order for this model to be useful, the author should include the data for 10 complementary base pairs (possibly also 11 base pairs), and clarify why the model didn't work. Are there some other limitations in this model?

We thank the referee for this remark. We noticed that the manuscript was not entirely clear with regard to the validity of the different levels of detail of the model and for the different regimes of high and low dCas9 concentration. As already outlined above we have now separated our conclusions more clearly into two parts: a) Passage of RNAP is possible even if dCas9 is saturating the target. b) dCas9 is fully or partially physically displaced from the target during successful transcription events.

The referee remarks that we don't demonstrate quantitative agreement of model and experiment down to 11 bp of complementarity for conditions where dCas9 does not saturate the target. First, we would like to stress that our first conclusion (transcription is possible while dCas9 is saturating the target) is supported by our experiments at high dCas9 concentrations for levels of complementarity from 11 to 20 bp. Then, we would like to stress that the data obtained for 10 bp of complementarity at low dCas9 expression are not incompatible with our kick-out model, but they are too noisy to make a quantitative comparison with the experiment. This is also already indicated in the manuscript (line 530: "We also carried out experiments with a guide RNA of 10 mismatches. However, the experimental data showed uncertainties too large to make conclusions about the level of α ."). There are multiple reasons for the increased uncertainty: For intermediate induction levels of dCas9 and 10bp complementarity the repression is very weak. Furthermore, as the P_{tet} promoter that is used to express dCas9 has a very steep activation curve, the intermediate induction can only be achieved in a very narrow range of aTc concentrations, leading to variability between replicates.

For levels of complementarity of 20 bp and 14 bp we demonstrate that the kick-out model fits our experimental results well if we assume a low rate of spontaneous or chromosome replication-induced unbinding (i.e., $\alpha = 0$ fits the data well). At the

same time, spontaneous unbinding events are still taken into account in the general form of the kick-out model (Eqs. 5 and 6). For levels of complementarity lower than 14 bp we still expect the general form of the kick-out model to work. In Fig 2b we show that dCas9 is saturating the target for 11 bp of complementarity. Thus, the probability of successful transcription in the presence of dCas9 is higher than at higher levels of complementarity, suggesting that kick-out events are occurring more frequently. At the same time, it is possible that spontaneous unbinding becomes more relevant, i.e., that the normalized spontaneous unbinding rate α becomes significantly larger than 0 for low dCas9 concentrations. Yet, we know from our experiments at high dCas9 concentrations (Fig 2b) that the combined rate of spontaneous and induced unbinding is not exceeding 10% of the rate of rebinding (95% confidence) for 11bp of complementarity.

In line 201 of the manuscript we now say that quantitative agreement between kick-out model and experiment is reached for high and intermediate (14 bp) complementarity and that we expect the general form of the model to work for low levels of complementarity for the reasons mentioned above, while spontaneous unbinding could become more important than the rate of ejections. Furthermore, we have explicitly mentioned the possibility that only a fraction of successful transcription events leads to dCas9 ejection.

4, The authors also came up with a model for the feedback loops. Interestingly, the model suggested that increased DAPG concentration strangely stabilizes dimerization of PhIF for repression. This needs to be verified by experiments to validate the model.

We thank the referee for this critical remark, and we agree that our data do not show the validity of the proposed stabilization of PhIF dimers. We have now stressed that dimer stabilization is only one of possibly multiple mechanisms how the sharp transition of Hill coefficients could be explained. Since validating the model by different means goes beyond the scope of our manuscript we have now removed this hypothesis from the main text and also added a sentence weakening our hypothesis in the appendix text. In the manuscript, we now write:

"To reconcile our observation we speculated that PhIF might be predominantly found as monomers at low DAPG concentrations and as dimers at high DAPG concentrations (see the Appendix for details). However, the detailed mechanism underlying the sharp transition in Hill coefficients remains to be studied by independent experiments."
(line 285)

In the Appendix text we write: "We note that this hypothesis remains highly speculative and other mechanisms might be responsible for the transition in Hill coefficients observed." (page 3, last paragraph).

5, In Figure 5, the author used cooperativity $n=1$ or $n=2$ for PhIF to fit the data in the feedback loops. However, $n=4.5$ was observed previously for PhIF. The authors need to address this discrepancy. In addition, it is not clear why $a=0.15$ was used in Figure EV6 to fit the model.

The Hill coefficient identified here describes the repression as a function of effective promoter strength, i.e., as a function of the gene expression in the absence of repressor. To our knowledge, this relationship has not been measured before. To avoid confusion, we have now added an explanation in the manuscript (line 267):

"To determine whether PhIF binds to the operator cooperatively, we aimed to quantify the feedback strength as a function of promoter strength for different DAPG

concentrations. To mimic different promoter strengths we targeted the *sfgfp* ORF using spacers with variable degrees of complementarity (Fig 5)."

A previously reported Hill coefficient of 4.7 (Stanton *et al*, 2014) describes the repression as a function of DAPG concentration, measured in a HEK293 human cell line. The same study also reports a Hill coefficient of 1.0 for the same repressor when measured in CHO cells, indicating a large variability of the response curve depending on the context. Our measurement in *E. coli* cells (as now explicitly indicated in Appendix Fig S10) gives a Hill coefficient of 1.25 ± 0.07 . While the characterization of the PhIF promoter as a function of DAPG concentration was not a focus of our work, we are also not aware of any previously published measurements of this Hill coefficient in *E. coli*.

Regarding the dimerization constant $a=0.15$: We have now stated explicitly in the Appendix text that the value was obtained empirically. We write in page 3, line 20: "Empirically, we found for a value of $a=0.15$ that the increasing PhIF concentration upon decreasing DAPG concentration can partially explain the transition in Hill coefficient (Appendix Fig S11c), while lower or higher values of a provide better fits for the regimes of low or high DAPG concentrations, respectively, but not for both regimes."

6. The interpretation of the Elowitz 2002 paper is often stretched to the point of being wrong. The comment regarding the expression noise difference between inducible and native promoters is not generally correct and this reference only looks at a few examples. The citation on lines 293-294 is also incorrect, but could be due to a grammatical problem or misplacement of the reference.

We agree with the referee that our statement was overly general. We now write: "Complementarity-based gene repression is qualitatively different from gene repression using transcriptional repressors. For example, the Lac repressor can increase the extrinsic part of the noise of its targets by about 5-fold as compared to the unrepresed case (see Appendix Fig S3 and also (Elowitz *et al*, 2002))."

Appendix Figure S3.

Regarding the citation Elowitz *et al.*, 2002 in line 352 of the manuscript: We think this is the right paper to cite when introducing the concept of extrinsic noise. Elowitz *et al.* show

in their Fig 3 that two fluorescent proteins expressed from the same promoter but in different sites on the chromosome show strong correlations in expression, just in the same way as mCherry and GFP in our experiments. However, if the referee thinks we should add additional references, we will of course consider those.

7. dCas9 can be very toxic when over expressed in the host. The ability to integrate it into the genome under inducible control is interesting and would be very useful. To what extent is there a growth impact at different levels of aTc induction? How stable is the strain when being passaged over multiple generations?

This is an excellent point to be considered when it comes to practical applications of the system. A new paragraph presenting experimental data on dCas9 toxicity was added to the results section (line 80) and Appendix Fig S1:

“Inducing dCas9 expression in this setup did not have an impact on growth (Appendix Fig S1). We also measured the stability of the target gene repression over time and saw repression over 5 days of culture while 40/40 of the clones tested recovered the target gene expression after we stopped dCas9 induction. This genetic system is thus very stable and dCas9 expression did not show any toxicity.”

Appendix Figure S1.

Reviewer #2

Vigouroux et al. reveal the underlying importance of the complementarity between the crRNA of CRISPR/Cas9 system to the target sequence for noiseless targeted gene repression. Their approach also leads to a new mechanistic model that accurately describes dCas9-mediated repression in bacteria. Importantly, the authors found that it is the extent of mismatch between the crRNA and the target gene that controls the degree of repression of the target gene independent of promoter strength, enabling fine-tuned controlled of (presumably) any gene expression. The authors also demonstrate that a "kick-out" model of dCas9 binding that does not rely on dCas9 concentration, but rather the degree of complementarity to the target sequence. This is an excellent study and I am supportive of publication assuming the following concerns are addressed.

We thank the reviewer for his positive assessment of our work.

1. It would be very useful to contrast these dCas9 results with alternative transcription factor-based control of the same gene. Specifically, can the authors reproduce Fig 1a-d using TALEs for repression and mutations in the target? Alternatively, LacI-LacO system with the known mutations in the LacO sequence that reduce the affinity.

We agree with the referee that it would be interesting to compare our approach with other proteins that target inside the gene. As long as a DNA-binding protein acts like a road block inside the ORF that can be kicked out by RNAP it should fulfill similar characteristics as our system. However, we have not attempt to do this, as it would require modifying the target sequence. That is apparently much harder from a practical viewpoint than modifying the sequence of the guide RNA, and it would potentially interfere with downstream translation.

To demonstrate the versatility of targeting the gene inside the ORF, we now contrast this approach with targeting the same sequence inside the promoter region. In this case, which is comparable to transcription-factor based approaches, we find that repression is stronger (as RNAP cannot kick out the dCas9 complex) and dCas9-concentration dependent (as transcription now relies on the rare times, where dCas9 is not bound). We now write (line 164):

"Interestingly, when the same target is moved from the ORF to the promoter region, repression is increased and depends on concentration of active dCas9 complex (Appendix Fig S5). This finding suggests that RNAP can pass the occupied target site inside the ORF thanks to its processive polymerase activity, but that dCas9 cannot bind at the occupied target site inside the promoter region, where it relies on diffusion."

2. Figure 5. Does not depict the various mismatch guides used, only the varying amount of DAPG.

This is now corrected.

3. Targeting the mrdAB operon for repression does not reduce copy number if the construct is integrated, i.e. not on a plasmid.

The *mrdAB* operon is in the native locus. By targeting the operon we change the amount of enzymes expressed. By "copy number fluctuations", we referred to fluctuations in the number of enzyme per cell. For better clarity, this term has been replaced by "amount" (abstract line 12) or "enzyme number" (manuscript line 306).

4. Figure 1a. What is the concentration of aTc for the various guides?

This information has been added to the main text and figure caption.

5. Figure 2. Decoy guides. How many active guides are in the cell? They integrated the array, with at least 2 guides for GFP and RFP, but only added one array. Important point because the authors claim that this approach can be multiplexed

This is a very good point. We now constructed arrays with up to 4 guides. The new results are now incorporated in the text (line 124):

"This remained true even with up to 4 spacers per array, regardless of the position of the active spacer in the array (Appendix Fig S4)." We have also included the following Appendix Fig S4, which shows our results for multiple guides and demonstrates that

multiple guides indeed lower the concentration of active dCas9 complexes (Appendix Fig S4b).

Appendix Figure S4.

Reviewer #3

Summary

This manuscript aims to show that endogenous genes can be repressed in microbial systems in a manner that does not increase transcriptional noise and is based on the degree of guide-RNA sequence complementarity, rather than repressor expression level. A consistent relative repression of a target gene can be generated by high (saturating) levels of dCas9 binding and blocking RNA polymerase. Repression strength is modulating by creating mismatches in the guide-RNA -- target DNA sequence that increases the probably that dCas9 will unbind when encountering a transcribing RNA

polymerase (the "kick-out" model). Further, temperature was shown to affect the kick-out rate. An established auto-regulatory feedback loop (PhIF) was used to demonstrate how repression under the kick-out model is dependent on transcription initiation rate. Additionally, this gene repression paradigm was used to knock-down two genes involved in cell shape and was shown to be able to modulate two promoters independently using guide-RNAs with different numbers of mismatches. Technically, the experiments were almost exclusively based on live, high-throughput imaging of the expression of fluorescent reporter proteins in populations of microbes.

General remarks

The data, as far as it goes, is convincing and the experiments are elegant and well-designed. I would recommend this manuscript for publication, however, some key controls and variables need to be included for CRISPR-Cas9 systems are becoming an important research tool for dissecting genetic mechanisms in biology. This manuscript demonstrates how inactive Cas9 systems can be used to repress genes with high controllability. The advances in this manuscript are both conceptual and technical in nature. The "kick-out" model of dCas9 unbinding is mechanistically interesting in that suggesting that differing guide-RNA mismatches is more controllable was to repress genes than changing repressor concentration. The technical advance is demonstrating the controllability of gene repression: consistent, relative fold-reductions in gene expression and a lack of increase in transcriptional noise introduced by the repression mechanism. This difference in transcriptional noise between repression based on guide-RNA mismatches and repressor concentration was a striking result and will be an important finding for the broader research community. The advance in fine-scale controllability of gene repression is a significant advance compared to more established methods of repression by increasing repressor concentration. This work will likely be implemented by laboratories interested in gene repression in microbial systems. All audiences interested in modulating gene expression levels in microbial systems would be highly interested in this work, including and especially synthetic biologists constructing gene circuits. Research in eukaryotes will also benefit from considering the data in this manuscript, although the situation is more complex in eukaryotes.

We thank the referee for these positive comments and for the encouraging words for the use of complementarity-based gene repression in bacteria and eukaryotes.

Major concerns

- A central conclusion of this manuscript is that at saturating conditions, the only determinant of relative repression is the degree of complementarity between guide-RNA and DNA (i.e. - the kick-out model). Figure 2 is purported to demonstrate this, however more extensive controls are needed for such a critical experiment. Specifically, in Figure 2a, a dummy guide is used to reduce active dCas9 levels, but only indirect evidence (the amount of processed guide-RNA) is provided and only for 2 conditions (a two-fold difference in active dCas9 level). Crucially, the amount of active-dCas9 should be directly measured through a dCas9-pull-down coupled to northern blot, quantitative-PCR, or sequencing. This can show that the targeting sequence and control sequence each comprise half of the dCas9-RNPs in the cell. Additionally, a more convincing demonstration of relative repression independence would include higher and lower levels of active dCas9 (until the point of sub-saturating conditions). This point impacts the interpretation of Figure EV3 as well (whether temperature is modulating repression strength via differential occupancy or $P(\text{stop})$).

We believe that the Northern blot assay we performed provides a reasonable estimate of the relative concentration of active complex as the processed crRNA is very unlikely to exist as a free species in any substantial amount. Performing a Pull down, as suggested by the reviewer, would have required techniques we do not currently master. Therefore this would have substantially delayed the resubmission of this manuscript. Nonetheless, we agree with the reviewer that dCas9-concentration independence is a crucial point of the study. We have therefore performed additional experiments to measure dCas9-mediated repression under a wider range of concentrations. Specifically we constructed CRISPR arrays containing up to 3 decoy spacers and we performed measurements at various aTc inducer concentrations to reach sub-saturating conditions. Indeed, we found that saturation is reached at higher aTc concentration in the case of the 3 decoys. These new results now provide excellent support for the conclusion that the concentration of active complex is saturating under the experimental conditions used in the rest of the study (100ng/ml of aTc). These new results are now shown in Appendix Fig S4b and described in the main text as follow (line 126):

“The effectiveness of the decoy strategy was confirmed by gradually lowering the concentration of aTc until we observed the transition from strong repression to no repression. As expected, the transition happened at higher aTc concentrations with three decoys than with one (Appendix Fig S4b), confirming that decoys reduce the concentration of active complex. In both cases, at high induction, the residual expression reached a plateau value around 3%, corresponding to the concentration-independent regime.”

- The effect of sequence content variability within the gRNA-target region should be addressed. This manuscript describes the effect of the number of mismatches, but not whether the identity of mismatching base-pairs effects the probability that dCas9 will stop transcription, $P(\text{stop})$. For example, does an eliminated G-C base pairing decrease $P(\text{stop})$ more than an eliminated A-T base-pairing in the complementary region due to its higher thermodynamic binding energy? This adds another dimension to the relative repression model and affects the generalizability of this paradigm for repressing arbitrary genes.

This is indeed an important question for the design of new CRISPR guides. We could observe in our own data that the expression level after repression (i.e. passage probability) cannot simply be predicted from the spacer complementarity, as seen in the following figure:

The construction of a model able to predict passage probability from sequence information and number of mismatch would require data collected in high-throughput screens designed to answer this question specifically. We believe that this goes beyond the scope of the present study. A paragraph on this topic has been added to the discussion (line 388):

“We explain these observations by a 'kick-out' model of repression, according to which RNAP kicks out dCas9 with a probability that can be tuned by spacer complementarity. The exact passage probability depends on the crRNA sequence. Here we provide a collection of guide RNAs against *mcherry* and *sfgfp* with known passage probabilities. Further work and larger datasets of diverse sequences will help to design new guides with predictable repression strength on arbitrary targets.”

- Figure 3 and EV2: Do these results extend to different promoters with different transcription rates? Only two have been tested and these differ in basal transcription rate by a factor of approximately 2-2.5. If these results are supposed to be generalizable to modulate the relative expression of arbitrary genes, a larger range of promoter strengths/transcription initiation rates should be tested as in Figure 3 (e.g. - 2-4 more promoters with higher and lower transcription rates).

The data that is presented in the paper compares $P_{\text{PHIF-gfp}}$ with $P_{127-gfp}$ (2.4-fold difference in transcription rate). We now also compared $P_{127-mCherry}$ to $P_{\text{Lac-mCherry}}$ with 1 mM of IPTG (up to 12-fold difference in expression), and found no difference in relative repression. The new data are now included in Appendix Figure S6 of the manuscript:

Appendix Figure S6.

In addition, we confirmed the independence on promoter strength at even lower transcription rate, by partially inducing the P_{Lac} promoter with different concentrations of IPTG and repressing it to various levels with CRISPR. Due to systematic pipetting error, the intermediate induction levels exhibited large variability between biological replicates, yet we could verify that on average the relative repression factor associated with one guide RNA was the same for any induction level. The data is presented below:

The lowest induction level used here gives an expression about 3 times lower than fully-induced P_{Lac}. Compared to Appendix Fig S6 and Fig 3, the transcription rate in that condition is presumably ~36 times weaker than P₁₂₇ and up to 150 times weaker than P_{PhIF} thus confirming that the transcription-rate independence remains true over a wide range of promoter strengths.

Minor points

- Related to Figure 1: A semantic issue, but possibly misleading to readers who do not

examine the raw data. The authors claim a "noise-less" repression is can be instantiated and controlled by the degree of complementarity between guide-RNA and target DNA. The data show a striking degree of noise introduced by repression based on dCas9 level, and that this extra noise is not generated by altering mismatch number, however the intrinsic biological noise of transcription is still present (as demonstrated by the overlapping distributions in Figure 1D. Therefore, it's more appropriate to refer to mismatch based repression as not adding additional noise to endogenous transcription rate, but the expression level is not noiseless.

This is a fair point. To clarify, the section title has been changed to:

"Varying levels of guide RNA-target complementarity enables controlling gene expression without addition of noise".

- *Figure 1f is not referenced or discussed in the text or supplement. This should be included.*
- *Figure 5: Missing labels. I'm guessing that the 5 data points on each line correspond to 20, 14, 11, 10, and 0-bp of complementarity (from left to right), but this must be included. Also, the figure says there is a "No Feedback" label on the graph (there is not), this corresponds to the "No PhIF" line, but the legend and figure should match.*
- *Line 145: Likely that figure 1d should be referenced instead of Figure 1c.*
- *Supplement: section "Quantitative comparison with experiment": Figure 2c is referenced, but this should be figure 3c.*

These four points have been corrected. We thank the reviewer for pointing them out.

- *Supplement: section "Kick-out model of CRISPR knock-down": It would be helpful to readers to explicitly derived Equation 2.4.*

We have now moved the model part to the Materials and Methods section. We have included a sentence preceding Eq. 2.4 (now Eq. 4 in Materials and Methods) (line 478): "Plugging Eq. (2) into Eq. (1) and using the Eq. (3) for k_{out} , the kick-out model of dCas9-based gene repression thus predicts a normalized transcription rate [...]"

Also related to the model, we have now slightly modified the model by including an ejection frequency δ in Eqs. (3, 4) that describes the fraction of successful collision events that lead to dCas9 ejection. The previous kick-out model and equilibrium unbinding models now correspond to $\delta = 1$ and $\delta = 0$, respectively (see line 459 following Eq. 3). However, any finite value of $\delta > 0$ is compatible with our experimental observations. We have therefore generalized the kick-out model to all cases $\delta > 0$. We have now included a brief discussion in the main text and in the Materials and Methods part.

- *Supplement: EV2: better figure labeled is needed to interpret this graph. Which are the predicted and experimental points? What is signified by the vertical lines?*

This figure (now Appendix Fig S7) has been re-designed for better readability. The vertical lines are now labeled. We have also substantially extended the figure legend.

- *Lines 267-269, Supplement EV8b: Considering how small the reduction in cell length was at high repression, I'm skeptical of the interpretation that the cell is minimizing the area:volume ratio to compensate for low PBP2. However, this does not negate the strong effect/conclusion of the cell morphology vignette of this manuscript.*

We agree with the referee that this conclusion is too strong and speculative. We have removed the corresponding sentence from the manuscript and simply write instead (line 324):

"[...], consistently with PBP2 and RodA being essential for building the cylindrical part of the cell wall but not the cell septum. We then wondered whether enzyme levels in individual cells were responsible for cell-to-cell variations in cell diameter at low or intermediate expression levels, where the average cell diameter was affected by *mrdAB* repression. Indeed, we found cell-to-cell fluctuations in the intracellular density of mCherry-PBP2 to be negatively correlated with cell diameter for intermediate *mrdAB* repression [...]."

- While not required for interpretation of the data in this manuscript, an interesting experiment would be to compare mismatches occurring at the 5'-end of the guide-RNA vs internal mismatches (that do not disrupt the 3'-end seed region). Does 6-bp of mismatch at the 5'-end cause a different P(stop) value than an internal 6-bp mismatch (a does it matter if the mismatches are adjacent, as one block, or interspersed throughout the target region)? A difference here would require a refinement of the conclusion that the number of mismatches determines relative repression level and would need to include mismatch location as a variable.

It is well described in the literature that mutations in the seed sequence (~8-12 nt PAM-proximal) have a very strong effect on Cas9 activity or dCas9 binding and were thus excluded from this study (Jinek *et al*, 2012; Boyle *et al*, 2017; Fu *et al*, 2013; Hsu *et al*, 2013; Jiang *et al*, 2013). The effect of mismatches in the PAM-distal region has also been investigated in various studies where it appears as a general rule that consecutive mismatches are more deleterious than spread-out mismatches, and the closer to the PAM the more deleterious (Boyle *et al*, 2017; Fu *et al*, 2013; Hsu *et al*, 2013). It is also known that binding in the PAM-distal region is required for Cas9 cleavage but not for binding (Bikard *et al*, 2013; Duan *et al*, 2014; Kuscu *et al*, 2014; Sternberg *et al*, 2015; Chen *et al*, 2017). The purpose of the present study being to propose and characterize a gene repression strategy we focused on a specific type of mutation: consecutive mismatches at the 5' end. This strategy provides a fine and broad range of dCas9 activity that perfectly suits our purpose. We agree that a more detailed investigation of the position and type of mismatches could reveal interesting aspects of the biophysics of dCas9 binding, but believe that this goes beyond the scope of the present work.

Additional references

- Bikard D, Jiang W, Samai P, Hochschild A, Zhang F & Marraffini LA (2013) Programmable repression and activation of bacterial gene expression using an engineered CRISPR-Cas system. *Nucleic Acids Res.* **41**: 7429–7437
- Boyle EA, Andreasson JOL, Chircus LM, Sternberg SH, Wu MJ, Guegler CK, Doudna JA & Greenleaf WJ (2017) High-throughput biochemical profiling reveals sequence determinants of dCas9 off-target binding and unbinding. *Proc. Natl. Acad. Sci. U. S. A.* **114**: 5461–5466
- Chen JS, Dagdas YS, Kleinstiver BP, Welch MM, Sousa AA, Harrington LB, Sternberg SH, Joung JK, Yildiz A & Doudna JA (2017) Enhanced proofreading governs CRISPR–Cas9 targeting accuracy. *Nature* **550**: 407
- Duan J, Lu G, Xie Z, Lou M, Luo J, Guo L & Zhang Y (2014) Genome-wide identification of CRISPR/Cas9 off-targets in human genome. *Cell Res.* **24**: 1009

Elowitz MB, Levine AJ, Siggia ED & Swain PS (2002) Stochastic Gene Expression in a Single Cell. *Science* **297**: 1183–1186

Fu Y, Foden JA, Khayter C, Maeder ML, Reyon D, Joung JK & Sander JD (2013) High-frequency off-target mutagenesis induced by CRISPR-Cas nucleases in human cells. *Nat. Biotechnol.* **31**: 822

Hsu PD, Scott DA, Weinstein JA, Ran FA, Konermann S, Agarwala V, Li Y, Fine EJ, Wu X, Shalem O, Cradick TJ, Marraffini LA, Bao G & Zhang F (2013) DNA targeting specificity of RNA-guided Cas9 nucleases. *Nat. Biotechnol.* **31**: 827

Jiang W, Bikard D, Cox D, Zhang F & Marraffini LA (2013) RNA-guided editing of bacterial genomes using CRISPR-Cas systems. *Nat. Biotechnol.* **31**: 233

Jinek M, Chylinski K, Fonfara I, Hauer M, Doudna JA & Charpentier E (2012) A Programmable Dual-RNA-Guided DNA Endonuclease in Adaptive Bacterial Immunity. *Science* **337**: 816–821

Kuscu C, Arslan S, Singh R, Thorpe J & Adli M (2014) Genome-wide analysis reveals characteristics of off-target sites bound by the Cas9 endonuclease. *Nat. Biotechnol.* **32**: 677

Stanton BC, Siciliano V, Ghodasara A, Wroblewska L, Clancy K, Trefzer AC, Chesnut JD, Weiss R & Voigt CA (2014) Systematic Transfer of Prokaryotic Sensors and Circuits to Mammalian Cells. *ACS Synth. Biol.* **3**: 880–891

Sternberg SH, LaFrance B, Kaplan M & Doudna JA (2015) Conformational control of DNA target cleavage by CRISPR–Cas9. *Nature* **527**: 110

Thank you for sending us your revised study. We are now satisfied with the modifications made and think that the study is suitable for publication.

Before we can formally accept your study for publication, we would ask you to address some remaining editorial issues listed below.

Corresponding Author Name: Sven van Teeffelen

Manuscript Number: MSB-17-7899